# K128 ubiquitination constrains RAS activity by expanding its binding interface with GAP proteins

Wout Magits [ID] [1,8], Mikhail Steklov[1,8], Hyunbum Jang[2], Raj N Sewduth [ID] [1,7], Amir Florentin [ID] [3], Benoit Lechat[1], Aidana Sheryazdanova [ID] [1], Mingzhen Zhang[2], Michal Simicek[4], Gali Prag [ID] [3], Ruth Nussinov [ID] [2,5] & Anna Sablina [ID] [1,6 ✉]

## Abstract

The RAS pathway is among the most frequently activated signaling nodes in cancer. However, the mechanisms that alter RAS activity in human pathologies are not entirely understood. The most prevalent post-translational modification within the GTPase core domain of NRAS and KRAS is ubiquitination at lysine 128 (K128), which is significantly decreased in cancer samples compared to normal tissue. Here, we found that K128 ubiquitination creates an additional binding interface for RAS GTPase-activating proteins (GAPs), NF1 and RASA1, thus increasing RAS binding to GAP proteins and promoting GAP-mediated GTP hydrolysis. Stimulation of cultured cancer cells with growth factors or cytokines transiently induces K128 ubiquitination and restricts the extent of wild-type RAS activation in a GAP-dependent manner. In *KRAS* mutant cells, K128 ubiquitination limits tumor growth by restricting RAL/ TBK1 signaling and negatively regulating the autocrine circuit induced by mutant KRAS. Reduction of K128 ubiquitination activates both wild-type and mutant RAS signaling and elicits a senescence-associated secretory phenotype, promoting RAS-driven pancreatic tumorigenesis.

**Keywords** Ubiquitination; RAS Signaling; RAS Interactome; NF1; Senescence-Associated Secretory Phenotype
**Subject Categories** Cancer; Post-translational Modifications & Proteolysis; Signal Transduction

## Introduction

Mutations of RAS proto-oncogenes (HRAS, NRAS, KRAS4A, and KRAS4B) are among the most common genetic alterations observed in human cancer (Punekar et al, 2022; Simanshu et al, 2017). The RAS pathway is also activated in a substantial subset of human pathologies in the absence of activating mutations of RAS. Nevertheless, the mechanisms leading to RAS hyperactivation in human pathologies have not been fully elucidated. Identifying these mechanisms could improve therapeutic approaches, block RAS activity, and facilitate the identification of patients who could benefit from RAS inhibitors.

RAS GTPases function as molecular switches alternating between an inactive GDP-bound and active GTP-bound state. The nucleotide-binding state of RAS is regulated by GTPase-activating proteins (GAPs, such as Neurofibromin 1 (NF1) and RAS P21 Protein Activator 1 (RASA1)) and guanine nucleotide exchange factors (GEFs, such as Son of Sevenless Homolog 1 (SOS1)). GTP-bound RAS activates a set of downstream effectors, such as RAF kinases, phosphatidylinositol 3-kinase (PI3K), and RAL guanine nucleotide dissociation stimulator (RalGDS) (Punekar et al, 2022; Simanshu et al, 2017). In addition to GEFs and GAPs, the activity of RAS GTPases is fine-tuned by posttranslational modifications (PTMs). The C-terminal farnesylation and/or prenylation, which anchor RAS to the membrane, were the first well-documented PTMs of RAS proteins. Further mass-spectrometry (MS)-based studies revealed additional PTMs of the GTP-binding domain of RAS protein, including phosphorylation, acetylation, and ubiquitination (Appendix Fig. S1A–C).

Ubiquitin serves as a versatile signal that may be conjugated as a single ubiquitin on a single lysine or multiple lysines of a target protein (monoubiquitination or multiubiquitination), or as poly-ubiquitination, in which lysines on the conjugated ubiquitin undergo multiple rounds of ubiquitination. The type and site of ubiquitin modification can alter RAS activity by controlling its degradation, subcellular localization, and interaction landscape (Campbell and Philips, 2021). Currently, understanding the mode of RAS ubiquitination, the functional consequences of ubiquitination at specific lysines, and its physiological relevance is a significant challenge.

Several ubiquitin E3 ligases, such as BTRC (β-TrCP), NEDD4, WDR76, SMURF2, and LZTR1/CUL3, have been implicated in the control of RAS stability by mediating its ubiquitination (Campbell and Philips, 2021). RABGEF1 (previously called RABEX5) has also

[1]VIB-KU Leuven Center for Cancer Biology, VIB, 3000 Leuven, Belgium. [2]Computational Structural Biology Section, Frederick National Laboratory for Cancer Research in the Laboratory of Cancer ImmunoMetabolism, National Cancer Institute, Frederick, MD 21702, USA. [3]School of Neurobiology, Biochemistry & Biophysics, The George S. Wise Faculty of Life Sciences, Tel Aviv University, Ramat Aviv, 69978 Tel Aviv, Israel. [4]Department of Hematooncology, University Hospital Ostrava, Ostrava, Czech Republic. [5]Department of Human Molecular Genetics and Biochemistry, Sackler School of Medicine, Tel Aviv University, Tel Aviv 69978, Israel. [6]Department of Oncology, KU Leuven, 3000 Leuven, Belgium. [7]Present address: Department of Oncology, KU Leuven, 3000 Leuven, Belgium. [8]These authors contributed equally: Wout Magits, Mikhail Steklov. ✉E-mail: anna.sablina@kuleuven.be

been shown to promote the mono- and di-ubiquitination of HRAS and NRAS, resulting in endosome recruitment and subsequent mitogen-activated protein kinase (MAPK) signaling impairment (Xu et al, 2010). On the other hand, the deubiquitinase OTUB1 inhibits the reversible ubiquitination of all RAS isoforms and promotes their localization to the plasma membrane, causing subsequent activation of MAPK signaling (Baietti et al, 2016). Although the contribution of specific ubiquitination sites to RAS degradation and endosomal localization is still unexplored, the role of site-specific ubiquitination on RAS activity is better described.

The monoubiquitination of KRAS at K147 leads to KRAS activation by impairing GAP-mediated hydrolysis and enhancing nucleotide-independent RAF binding (Baker et al, 2013a; Sasaki et al, 2011). The ubiquitin conjugation at lysine 117 (K117) allosterically promotes nucleotide exchange and accelerates GTP loading (Baker et al, 2013b). Ubiquitination at K104, which is spatially distant from the KRAS active site or effector binding region, would favor a conformation that facilitates the interaction of KRAS with the GEF protein SOS1, promoting RAS activation (Yin et al, 2020). On the other hand, monoubiquitination at the C-terminus impacts the tethering of HRAS proteins to the membrane, as ubiquitin conjugation results in the burial of the farnesyl and palmitoyl groups by ubiquitin, detaching HRAS from the membrane and impeding its downstream signaling (Steklov et al, 2018). Even though ubiquitination at K128 is the most frequently detected posttranslational modification of the G-domain of KRAS and NRAS (Appendix Fig. S1A–C), it is still unclear how this modification could affect RAS activity and function. In this study, we functionally characterized the role of RAS ubiquitination at K128 in controlling RAS signaling in both wild-type and mutant RAS models.

# Results

## Monoubiquitination at K128, the most prevalent modification of NRAS and KRAS, does not affect their stability and subcellular localization

Mass-spectrometry (MS)-based ubiquitome studies (Appendix Fig. S1A–C) predominantly utilize diGly-antibody enrichment that presents several limitations. For example, diGly-remnant peptides could derive from other ubiquitin-like modifications, such as ISGylation or SUMOylation, as well as present a bias toward specific amino acid sequences of remnant peptides (Ovaa and Vertegaal, 2018). To overcome these limitations, we used a tandem affinity purification approach to identify the ubiquitination sites of NRAS and KRAS in an unbiased manner. In line with previous studies, immunoblot analysis confirmed that both NRAS and KRAS are predominantly found in mono- and diubiquitinated forms.

MS analysis revealed ubiquitination at lysines 104, 117, 128, and 147 for both NRAS and KRAS; and lysine 135 in NRAS (R135 in KRAS) (Fig. 1A; Appendix Fig. S1D). The spatial analysis of the identified sites on the 3D structures of RAS proteins revealed two clusters of ubiquitination, located on the GTP-binding pocket (K117 and K147) and the α4 helix of the RAS protein (K128 and K135) (Fig. 1B). We observed a different distribution of ubiquitination at K117 and K147 for NRAS and KRAS, even though we identified multiple peptides corresponding to these sites

(Appendix Fig. S1D). On the other hand, the ubiquitination at K128 was the most prevalent posttranslational modification detected for either NRAS or KRAS proteins (Fig. 1A; Appendix Fig. S1D), in accordance with the data from Phosphosite (Appendix Fig. S1A–C). In HRAS, these sites were not modified as arginine residues are present at both positions 128 and 135 (Appendix Fig. S1C). In line with these findings, the K128R mutation led to a dramatic decrease in NRAS ubiquitination, indicating that K128 is the major site of NRAS ubiquitination (Fig. 1A; Appendix Fig. S1D).

We next investigated how K128 ubiquitination might regulate KRAS and NRAS. Since K135 is also located in proximity to K128 within the α4 helix (Fig. 1B), it is likely that NRAS monoubiquitination at K135 could also similarly affect RAS function. Therefore, to examine the role of ubiquitination of the α4 helix of NRAS, we used a double K128R/ K135R mutant. Multiple studies demonstrated the contribution of the ubiquitin system to the control of RAS protein stability (Campbell and Philips, 2021). Thus, we first assessed the effect of K128 ubiquitination on RAS stability using a global protein stability (GPS) approach. We generated a RAS stability reporter system that contains DsRed as a control to normalize protein synthesis, followed by an internal ribosome entry site sequence followed by the sequence coding for GFP-fused wild-type (wt)-RAS or ubiquitination-deficient mutants (Najm et al, 2021). The GPS assay revealed that both KRAS-K128R and NRAS-K128R/ K135R mutant showed a similar level of protein stability compared to wild-type proteins (Fig. 1C), indicating a non-degradative role of K128 ubiquitination for both KRAS and NRAS.

Several reports also demonstrated that mono- and diubiquitination of either NRAS or KRAS can lead to the endosomal association of RAS proteins (Baietti et al, 2016; Jura et al, 2006). However, we observed a similar distribution pattern for either wild-type or ubiquitination-deficient K128R or K128R/ K135R mutants (Fig. 1D), indicating that the ubiquitination at K128 does not affect the subcellular localization of NRAS and KRAS.

## Monoubiquitination of the RAS GTPases at K128 facilitates the binding of GAP proteins

Recent studies have shown that the monoubiquitination of small GTPases could determine their interaction landscapes (Sapmaz et al, 2019; Sewduth et al, 2023; Shin et al, 2017; Simicek et al, 2013). To explore this idea, we performed in silico modeling by attaching ubiquitin to the K128 residue of NRAS. We generated six models for ubiquitinated NRAS, covering a large spectrum of potential orientations, excluding transient contacts of ubiquitin with NRAS (Appendix Fig. S2A). During these simulations, we observed that the ubiquitin molecule converged to practically the same region of NRAS for all tested models. Clustering of the conformations across the simulations showed that the ubiquitin mainly interacted with the α3 and α4 helices on the allosteric lobe of NRAS (Appendix Fig. S2B).

To assess the effect of monoubiquitination at K128 on the RAS interactions, we aligned common RAS interactors to the simulated ubiquitinated NRAS complex by superimposing the structures of K128-ubiquitinated NRAS with RAS bound to its interactor proteins. We observed no additional interaction between ubiquitin and the RAS binding domain (RBD) of RAF1, which binds to the

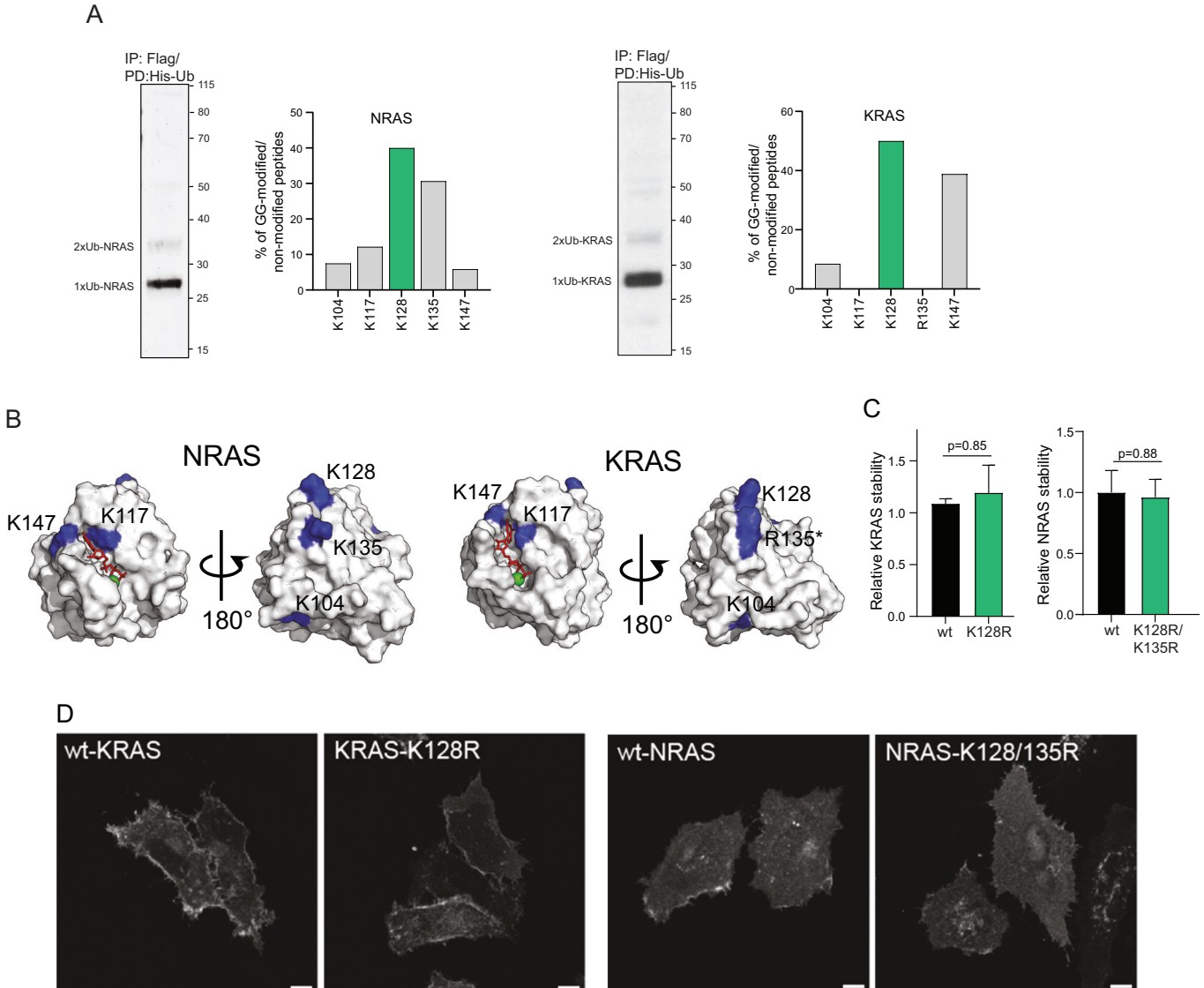

**Figure 1. K128 is the major site of RAS ubiquitination.**

(**A**) Tandem affinity purification of ubiquitinated NRAS and KRAS. 6xHis–ubiquitin and Flag-NRAS or KRAS were co-transfected into HEK293T cells, and ubiquitinated RAS proteins were purified using anti-Flag resin followed by $Co^{2+}$ metal affinity chromatography. Isolated RAS protein was visualized by immunoblotting with an anti-RAS antibody. Percentage of diGly-modified peptides to the total of identified peptides. $N = 3$. (**B**) Location of the identified ubiquitination sites on the 3D structures of RAS proteins. (**C**) Stability of the indicated proteins using the global-protein stability (GPS) approach. RAS turnover was monitored in HEK293T cells by FACS analysis. Data were shown as mean ± s.e.m, $N = 4$–5 technical replicates in independent experiments. *P*-value was determined by a two-sided *t*-test. (**D**) Immunofluorescence analysis of the indicated HA-tagged RAS proteins overexpressed in HeLa cells. Scale bar, 10 µm. Source data are available online for this figure.

effector lobe of NRAS opposite to K128 (Appendix Fig. S2C). Similarly, the placement of K128-ubiquitinated NRAS into the NRAS/ SOS1 complex presented limited overlap (Appendix Fig. S2D). In contrast, a large contact interface area was detected between ubiquitin and the GAP domain of RASA1 (RASA1$^{GAP}$) (Appendix Fig. S2E), suggesting that K128-ubiquitinated NRAS could exhibit an additional interaction interface with RASA1$^{GAP}$ when compared to non-ubiquitinated NRAS. To further characterize the predicted interaction of RAS-conjugated ubiquitin with RASA1, we performed molecular dynamics (MD) simulations on K128-ubiquitinated NRAS/ RASA1$^{GAP}$ complex in an aqueous environment. The most representative complex configurations

from the clustering of the simulation ensembles revealed that the ubiquitin had a high probability of contacting RASA1$^{GAP}$ (Fig. 2A). In silico modeling further indicated that the ubiquitination of NRAS at K128 strengthened its interaction with RASA1$^{GAP}$ (Fig. 2B).

To validate these in silico predictions, we chemically attached ubiquitin to NRAS at position 128 (Appendix Fig. S3A), as previous studies confirmed that this approach mimics native ubiquitin conjugation (Baker et al, 2013a). Specifically, we substituted the residues K128 of NRAS and G76 of ubiquitin for cysteine residues to replace the native isopeptide linkage with a disulfide bond between C128 of NRAS and C76 of ubiquitin. We also mutated a

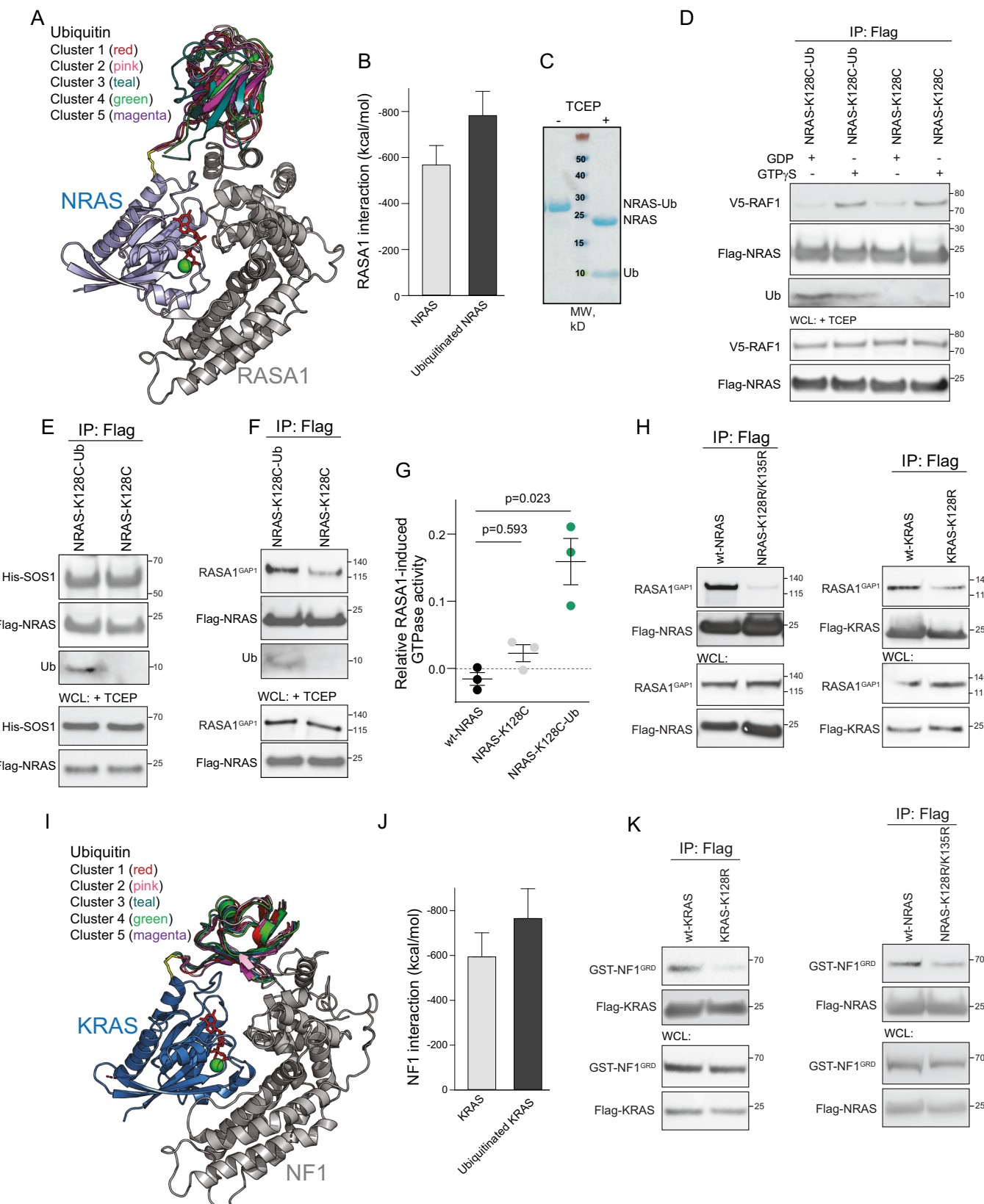

Figure 2. Ubiquitination at K128 facilitates the interaction of RAS with the GAP proteins.

(A) Superimposition of five representative complex conformations from the ensemble clusters for the K128-ubiquitinated NRAS/ RASA1$^{GAP}$ complex. (B) The interaction energies of RASA1$^{GAP}$ with non-ubiquitinated NRAS and NRAS ubiquitinated at K128. The interaction energies were calculated over the entire trajectories and then averaged. Data represented as mean ± SD. $N = 1250$ snapshots. (C) Chemical ubiquitination of NRAS-C118S/ K128C mutant and purification of monoubiquitinated NRAS by size-exclusion chromatography. Proteins were separated by SDS-PAGE under non-reducing conditions or in the presence of a reducing agent, tris(2-carboxyethyl) phosphine (TCEP), and stained by Coomassie blue. (D) In vitro Co-IP of non-conjugated or ubiquitin-conjugated NRAS-K128C and RAF1. Equimolar amounts of NRAS-K128C/ Ubiquitin-G76C complex or NRAS-K128C mutant were incubated with V5-tagged RAF1 in the presence of GDP or GTPγS. (E) In vitro Co-IP of non-conjugated or ubiquitin-conjugated NRAS-K128C and SOS1. Equimolar amounts of NRAS-K128C-Ubiquitin-G76C complex or NRAS-K128C mutant were incubated with equal amounts of the GDP/GTP exchange domain of SOS1 in the presence of GDP. (F) In vitro Co-IP of non-conjugated or ubiquitin-conjugated NRAS-K128C and RASA1. Equimolar amounts of the indicated NRAS proteins were incubated with equal amounts of RASA1$^{GAP}$ in the presence of GTPγS. (G) RASA1-mediated GTPase activity of ubiquitinated and non-ubiquitinated NRAS normalized to intrinsic GTPase activity. Equimolar amounts of the indicated NRAS proteins were mixed with the GAP domain of RASA1. GTP hydrolysis reaction was initiated by the addition of GTP. Data were present as mean ± s.e.m. $N = 3$ technical replicates in independent experiments. P-value was calculated by a two-sided t-test. (H) wt-RAS and ubiquitination-deficient RAS mutant were co-immunoprecipitated with GST-tagged RASA1$^{GAP}$ using anti-Flag resin followed by immunoblotting with the indicated antibodies. (I) Superimposition of five representative complex conformations from the ensemble clusters for the K128-ubiquitinated KRAS/ NF1$^{GRD}$ complex. (J) The interaction energies of NF1$^{GRD}$ with non-ubiquitinated KRAS and KRAS ubiquitinated at K128. Data represented as mean ± SD. $N = 1250$ snapshots. (K) The indicated RAS proteins were co-immunoprecipitated with GST-tagged NF1$^{GRD}$ using anti-Flag resin, followed by immunoblotting with the indicated antibodies. Source data are available online for this figure.

surface-accessible residue of NRAS, C118, to serine to avoid additional chemical ubiquitination on this site. As a previous report demonstrated that the C118S mutation did not alter the RAS structure or its biochemical properties (Hobbs et al, 2013; Williams et al, 2003), we performed ubiquitin modification of NRAS at position 128 by adding a fivefold excess of ubiquitin-G76C. Monoubiquitinated NRAS was then purified from the reaction mixture through two consecutive rounds of size-exclusion chromatography (Fig. 2C; Appendix Fig. S3B).

In pull-down experiments, K128-monoubiquitinated NRAS loaded with a non-hydrolyzable GTPγS analog bound to RAF1 in a comparable manner to non-ubiquitinated NRAS (Fig. 2D), confirming that the monoubiquitination at residue K128 did not affect the interaction between NRAS and RAF1. Ubiquitin conjugation also did not affect the binding of GDP-bound NRAS to the GEF protein, SOS1 (Fig. 2E). In contrast, ubiquitin conjugation at K128 increased the binding affinity of GTPγS-bound NRAS with RASA1 (Fig. 2F). Moreover, we found that NRAS, when chemically ubiquitinated at position 128, showed higher RASA1-mediated GTPase activity when compared to non-ubiquitinated NRAS-K128C mutant (Fig. 2G), indicating that the increased interaction of K128-monoubiquitinated NRAS with RASA1 potentiates RASA1-mediated GTPase activity of NRAS.

Since K135 is also located within the α4 helix in proximity to K128 (Fig. 1B), it is likely that NRAS monoubiquitination at K135 could similarly affect the interaction between NRAS and RASA1. Therefore, to examine the role of ubiquitination of the α4 helix of NRAS, we utilized a double K128R/ K135R mutant. We observed that the double ubiquitination-deficient K128R/ K135R mutant of NRAS exhibited decreased binding to RASA1$^{GAP}$ compared to wt-NRAS (Fig. 2H). Similarly, the K128R mutant of KRAS showed decreased binding to RASA1$^{GAP}$ in comparison to wt-KRAS (Fig. 2H). These findings collectively indicate that the mono-ubiquitination of RAS proteins at K128 increases their interaction with the GAP protein RASA1, potentiating RASA1-mediated GTPase activity.

We next examined whether the ubiquitination at K128 could affect the interaction of RAS proteins with other GAP proteins, such as NF1. Modeling using cryoelectron microscopy (cryo-EM) structures of the full-length NF1 homodimer (Chaker-Margot et al, 2022), revealed that the GAP-related domain of NF1 (NF1$^{GRD}$) is exposed for RAS binding only in open conformation. The modeling of K128-ubiquitinated KRAS with the open conformation of the NF1 homodimer further showed that ubiquitin could interact only with NF1$^{GRD}$ (Appendix Fig. S4). Therefore, we performed MD simulations on the K128-ubiquitinated KRAS in a complex with the NF1$^{GRD}$ (Appendix Fig. S5). In the simulations, while KRAS conformation was stable, both ubiquitin and NF1$^{GRD}$ displayed conformational changes to accommodate the RAS-NF1 interactions. The most representative complex conformations from the clusters revealed that KRAS-bound ubiquitin showed a high probability of contacting the NF1$^{GRD}$, which strengthened its interaction with KRAS (Fig. 2I,J).

In line with the results of in silico modeling, we found that either ubiquitination-deficient K128R mutant of KRAS or K128R/ K135R mutant of NRAS showed decreased binding to NF1$^{GRD}$ compared to wild-type proteins (Fig. 2K). Collectively, these observations indicate that the monoubiquitination of RAS proteins at K128 can augment the interaction with the GAP proteins NF1 and RASA1.

## Ubiquitin-binding properties of GAP proteins

In silico modeling further revealed the formation of an interface between ubiquitin and GAP domains of either RASA1 or NF1 (Fig. 2A, I), suggesting that this interface may contribute to a stronger affinity of K128-ubiquitinated RAS to the GAP proteins in comparison to non-ubiquitinated RAS. For the NRAS/ RASA1$^{GAP}$ complex, the analysis predicted the presence of two, highly probable salt bridges, E34$^{ub}$-R1016$^{RASA1}$ and R74$^{ub}$-E1015$^{RASA1}$, that likely support the interaction between ubiquitin and RASA1. In addition to these salt bridges, two transient salt bridges, E34$^{ub}$-K1024$^{RASA1}$ and R74$^{ub}$-E1008$^{RASA1}$ might further favor the interaction of ubiquitin with RASA1 (Fig. 3A). Altogether, this indicates that the GAP domain of RASA1 presents ubiquitin-binding properties.

To validate the prediction models, we assessed the ability of the GAP domain of RASA1 to bind ubiquitin. Using a bacterial Split-Chloramphenicol Acetyl Transferase (Split-CAT) assay (Levin-Kravets et al, 2021), we detected the binding of wt-RASA1$^{GAP}$ to ubiquitin, whereas E1015A/ R1016A mutations impaired ubiquitin binding (Fig. 3B). Moreover, in contrast to wt-RASA1$^{GAP}$, the

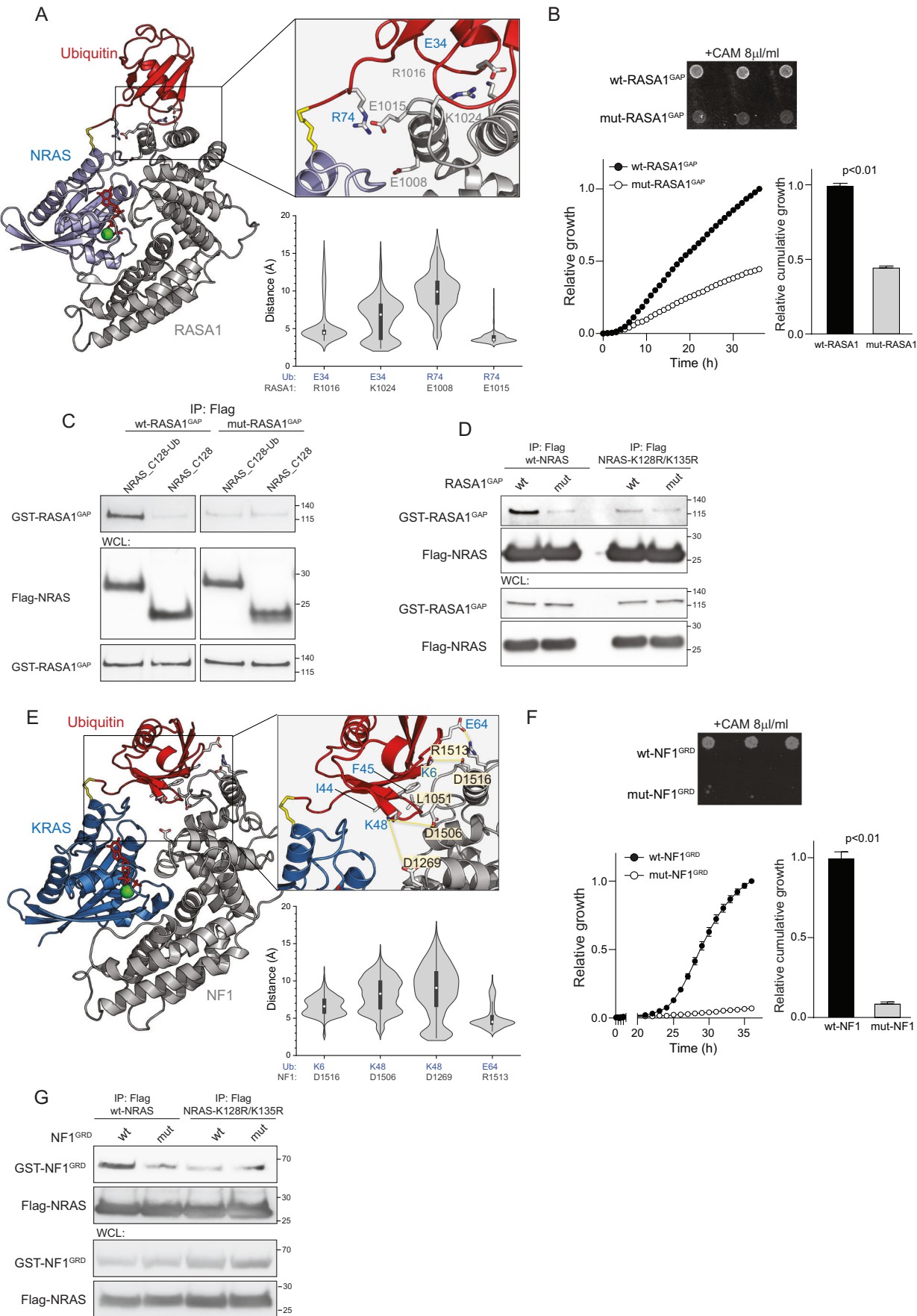

**Figure 3. The GAP proteins interact with ubiquitin.**

(A) The best representative complex conformation (cluster 1) from the ensemble clusters for the K128-ubiquitinated NRAS/ RASA1$^{GAP}$ complex, highlighting the salt bridges. The distance of four salt bridges was measured. Data were represented by a density plot with the median as the center, the interquartile range indicated with a rectangular box and the minimum/maximum value as endpoints. The density reflects the frequency distribution. $N = 625–1000$ snapshots. (B) The interaction between RASA1$^{GAP}$ and Ubiquitin using Split-CAT-based binding assay. Ubiquitin fused to C-CAT fragment and wt-RASA1$^{GAP}$ or the RASA1$^{GAP}$-E1015A/ R1016A fused to N-CAT were co-expressed in *E. coli*. The assembly of the N- and C-CAT fragments was monitored by *E. coli* growth in the presence of chloramphenicol. Data were present as mean ± s.e.m. $N = 5$ technical replicates in independent experiments. *P*-value was calculated by two-way ANOVA. (C) In vitro Co-IP of non-conjugated or ubiquitin-conjugated NRAS-K128C with either wt-RASA1$^{GAP}$ or RASA1$^{GAP}$-E1015A/ R1016A mutant. (D) The indicated NRAS proteins were co-immunoprecipitated with GST-tagged wt-RASA1$^{GAP}$ or RASA1$^{GAP}$-E1015A/ R1016A mutant using anti-Flag resin followed by immunoblotting with the indicated antibodies. (E) The best representative complex conformation (cluster 1) from the ensemble clusters for the K128-ubiquitinated KRAS/ NF1$^{GRD}$ complex, highlighting the salt bridges. The ubiquitin generated an extensive interface with NF1$^{GRD}$. The distance of four salt bridges was measured. Data were represented by a density plot with the median as a center, the interquartile range indicated with a rectangular box and the minimum/maximum value as endpoints. The density reflects the frequency distribution. $N = 1606$ snapshots. (F) The interaction between NF1$^{GRD}$ and Ubiquitin using Split-CAT-based binding assay. Ubiquitin fused to C-CAT fragment and wt-NF1$^{GRD}$ or NF1$^{GRD}$-L1501A/ D1506A/ R1513A mutant fused to N-CAT were constitutively co-expressed in *E. coli*. The assembly of the N- and C-CAT fragments was monitored by the *E. coli* growth in the presence of chloramphenicol. Data were present as mean ± s.e.m. $N = 5$ technical replicates in independent experiments. *P*-value was calculated by two-way ANOVA. (G) The indicated NRAS proteins were co-immunoprecipitated with GST-tagged wt-NF1$^{GRD}$ or NF1$^{GRD}$- L1501A/D1506A/R1513A mutant using anti-Flag resin followed by immunoblotting with the indicated antibodies. Source data are available online for this figure.

RASA1$^{GAP}$-E1015A/ R1016A mutant showed similar binding affinities to either chemically ubiquitinated or non-ubiquitinated NRAS proteins (Fig. 3C). The E1015A/ R1016A mutations of RASA1$^{GAP}$ also abolished the difference in its binding to wild-type or ubiquitination-deficient RAS proteins (Fig. 3D). Altogether, these results indicate that the interaction of RASA1 with ubiquitin through the aforementioned salt bridges would likely facilitate its binding to ubiquitinated NRAS.

The analysis of the K128-ubiquitinated KRAS/ NF1$^{GRD}$ complex also revealed that the interaction between ubiquitin and NF1$^{GRD}$ was stabilized through a highly probable salt bridge E64$^{ub}$-R1513$^{NF1}$ and three transient salt bridges K6$^{ub}$-D1515$^{NF1}$, K48$^{ub}$-D1506$^{NF1}$, and K48$^{ub}$-D1269$^{NF1}$ (Fig. 3E). In addition, the hydrophobic interactions between I44$^{ub}$/ F45$^{ub}$ and L1501$^{NF1}$ enhanced the interaction of ubiquitin with the NF1$^{GRD}$. The Split-CAT assay further confirmed the interaction between the wild-type GAP domain of NF1 and ubiquitin, whereas L1501A/ D1506A/ R1513A mutations of NF1$^{GRD}$ disrupted the ubiquitin-binding ability of NF1 (Fig. 3F). In line with these results, immunoprecipitation experiments showed that the NF1$^{GRD}$ mutant bound to either wild-type or ubiquitination-deficient NRAS in a similar way (Fig. 3G). Taken together, the non-covalent interactions of GAP proteins, RASA1 and NF1, with ubiquitin assists their recruitment to ubiquitinated NRAS and KRAS proteins.

## Ubiquitination at K128 constrains serum-stimulated activation of RAS proteins

Recent studies showed increased levels of KRAS ubiquitination at K128 in response to CD4$^+$ T-cell receptor stimulation or B-cell receptor stimulation (Dybas et al, 2019; Satpathy et al, 2015). More specifically, stimulation of A20 cells using an IgG antibody led to a transient increase in NRAS and KRAS ubiquitination at K128, whereas ubiquitination at K147 showed only a slight increase, and K117 was not affected (Appendix Fig. S6). Concordantly, the ubiquitome analysis of serum-stimulated murine embryonic fibroblasts showed a transient increase in KRAS and NRAS ubiquitination at K128, 5 min after serum stimulation (Fig. 4A). Immunoblotting analysis of serum-stimulated HeLa cells also demonstrated an increase of mono- and diubiquitination of NRAS, 5 min after serum stimulation (Fig. 4B). Moreover, it has been

reported that oncogenic RAS mutants show higher levels of ubiquitination compared to wild-type protein (Xu et al, 2010). This suggests that the ubiquitination of RAS is linked to its activation. Given that K128 ubiquitination of RAS proteins facilitates their GAP-mediated GTPase activity, a transient increase of RAS ubiquitination in response to growth factors and/or cytokines suggests that K128 ubiquitination contributes to the negative feedback loop restricting RAS activation upon stimulation.

To assess this idea, we generated conditional knock-in cells expressing either wild-type or ubiquitination-deficient mutants of *KRAS* or *NRAS*. By using the CRISPR-Cas9 technology, we introduced a cassette containing both wild-type and K128R or K128R/ K135R exons into the endogenous locus of either *KRAS* or *NRAS* and selected single-cell clones containing the cassette (Fig. 4C; Appendix Fig. S7). Using the *Cre-LoxP* system, the expression of wild-type RAS could be switched to K128R-mutant RAS after genetic recombination. RT-qPCR with primers specific to wild-type *KRAS* or *NRAS* confirmed the removal of the wild-type allele after adenoviral Cre-recombinase expression (Fig. 4D). This system allowed us to overcome the issue of single-clone heterogeneity.

We observed increased levels of phosphorylated MEK1/ 2 and ERK1/ 2 in HeLa cells expressing K128 ubiquitination-deficient RAS mutants, specifically 5 min after serum stimulation (Fig. 4E). This suggests that ubiquitination at K128 reverses serum-induced activation of NRAS and KRAS. Depletion of either *NF1* or *RASA1* expression with shRNA abolished the changes observed between cells expressing wild-type RAS or ubiquitination-deficient mutants (Fig. 4F; Appendix Fig. S8), indicating that K128 ubiquitination-mediated downregulation of the RAS-MAPK pathway is GAP-dependent. Thus, the K128 ubiquitination of RAS induces a GAP-dependent negative regulatory loop, restricting RAS activity. Notably, the ubiquitome analysis of the Clinical Proteomic Tumor Analysis Consortium (CPTAC) lung squamous cell carcinoma (LUSC) dataset (Satpathy et al, 2021) revealed that lung tumor samples presented decreased levels of NRAS and KRAS ubiquitination at K128 in comparison to normal tissue (Fig. 4G). On the other hand, ubiquitination of KRAS at K117 was not changed in tumor samples, whereas ubiquitination of RAS at other sites was not detected in the CPTAC ubiquitome analysis (Appendix Fig. S6B). This suggests that the dysregulation of the ubiquitination of RAS at

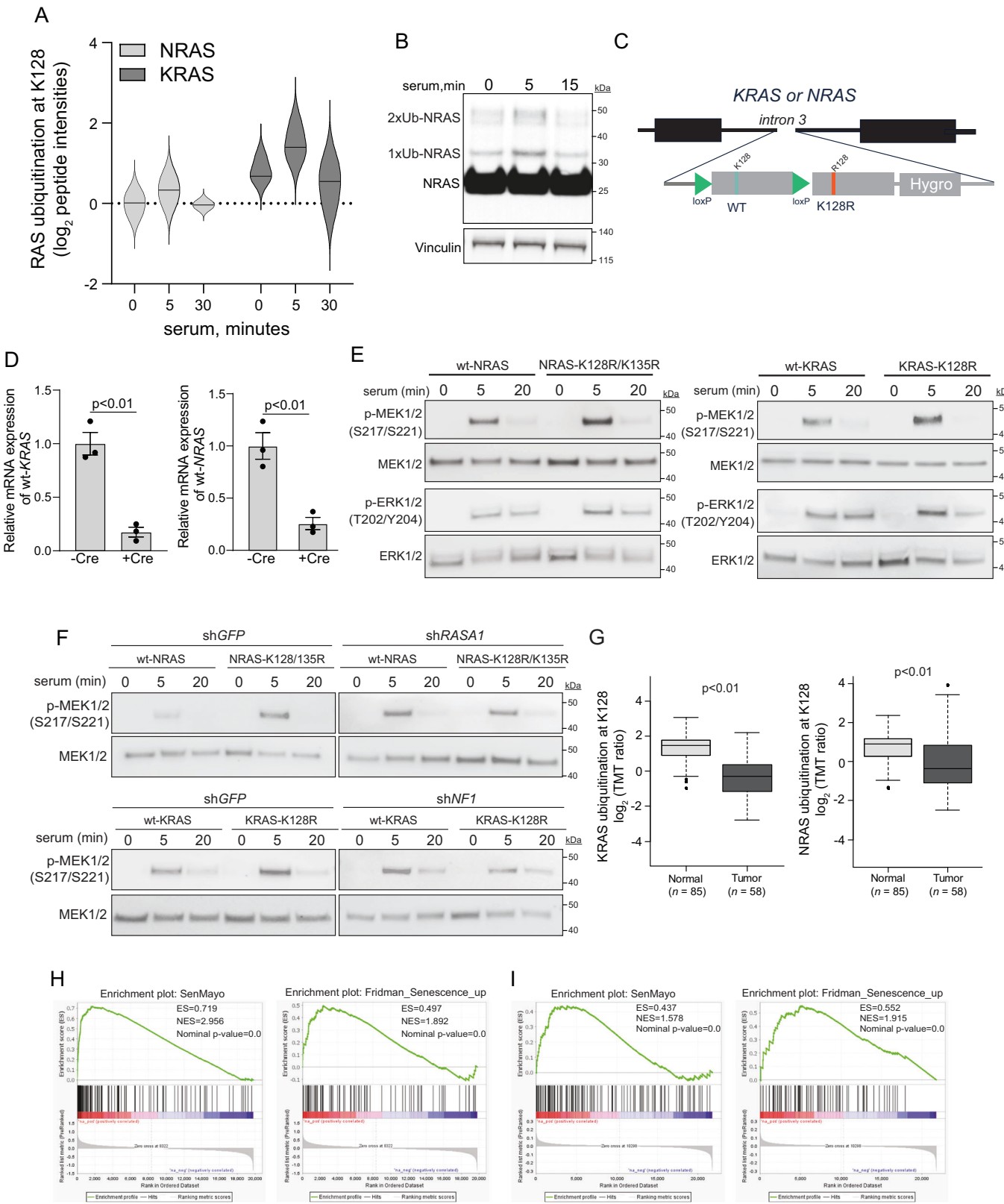

**Figure 4. Ubiquitination at K128 restricts the extent of wild-type RAS activation.**

(A) K128 ubiquitination of NRAS or KRAS detected by the MS-based ubiquitinome analysis upon 10% serum stimulation of serum-starved MEFs for the indicated time periods. $N = 3$ biological replicates. (B) Immunoblot analysis of NRAS ubiquitination. HeLa cells were co-transfected with Flag-tagged NRAS and ubiquitin. Twenty-four hours after transfection, cells were serum-starved overnight and then stimulated for the indicated time. (C) Schematic illustration of the CRISPR-Cas9-based approach to generate conditional K128R knock-in HeLa clones. (D) Expression of wt-RAS in the single-cell clones harboring the RAS knock-in cassette after overexpression of Cre recombinase. Data were shown as mean ± s.e.m. *P*-value is determined by a two-sided *t*-test. $N = 3$ technical replicates in independent experiments. (E, F) The indicated HeLa cells were serum-starved overnight, stimulated with 10% serum, and analyzed by immunoblotting. (G) K128 ubiquitination of RAS proteins detected by CPTAC ubiquitinome analysis in tumor ($n = 85$) and normal ($n = 58$) tissue of LUSC patients. Data were represented as box plots with the median as the center, the interquartile range (IQR) indicated with a rectangular box, and the whiskers are defined by the first and third quartile ± 1.5x IQR. Data were shown as mean ± s.e.m. *P*-value is determined by a two-sided *t*-test. (H) GSEA analysis of the TCGA LUSC tumors stratified by *NF1/ RASA1* status. Statistical analysis was performed by permutation test. (I) GSEA analysis of the CPTAC LUSC tumors stratified by the levels of KRAS ubiquitination at K128. Statistical analysis was performed by permutation test. Source data are available online for this figure.

K128 contributes to a feedback loop important for lung cancer development and progression.

The loss of the RAS GAPs has also been linked to oncogene-induced senescence (Courtois-Cox et al, 2006). Concordantly, the gene set enrichment analysis (GSEA) of the TCGA data from LUSC patients revealed that loss of *NF1* or *RASA1* was associated with senescence expression signatures, SenMayo (Saul et al, 2022) and Fridman_Senescence_up (Fridman and Tainsky, 2008) (Fig. 4H). This suggests that dysregulation of RAS ubiquitination at K128 could also lead to senescence-related phenotypes. In agreement with this idea, treatment with the neddylation inhibitor MLN4924, which blocks RAS ubiquitination (Steklov et al, 2018), leads to cellular senescence (Ni et al, 2022). Moreover, we found that LUSCs with low levels of KRAS ubiquitination at K128 showed significant enrichment for the senescence signatures (Fig. 4I). These results suggest the potential role of RAS ubiquitination in controlling senescence-related phenotypes.

## Dysregulation of K128 ubiquitination of KRAS-G12D promotes tumorigenesis by inducing the senescence-associated secretory phenotype

Recent studies reported that *NF1* loss accelerates tumorigenesis driven by mutant KRAS, indicating that wild-type NF1 might restrain oncogenic KRAS signaling (Ramakrishnan et al, 2022; Wang et al, 2019). Given that the observed effects of K128 ubiquitination are NF1-dependent, we hypothesized that K128 ubiquitination could affect mutant KRAS signaling. Thus, we introduced a conditional knock-in for KRAS-K128R mutation in the pancreatic cancer cell line SW1990 harboring a homozygous *KRAS-G12D* mutation (Fig. 5A; Appendix Figs. S7 and S9A). We found that introducing the K128R mutation in KRAS-G12D led to an increase in 2D colony growth in all three independent SW1990 clones (Fig. 5B). Moreover, K128R knock-in led to an increase in the SW1990 xenograft growth (Fig. 5C,D).

The analysis of RAS signaling revealed that K128R knock-in did not lead to consistent alteration of MAPK or AKT activity in different clones (Appendix Fig. S9B). On the other hand, the K128R mutation increased levels of Tank-Binding Kinase 1 (TBK1) activation in all three single-cell clones (Fig. 5E). Notably, levels of TBK1 phosphorylation correlated with the Cre recombination efficiency in the individual clones (Fig. 5E; Appendix Fig. S9A), further implying that K128R mutation leads to TBK1 activation. SW1990-KRAS-G12D/ K128R xenografts also showed increased levels of phosphorylated TBK1 (Fig. 5F). Therefore, the activation

of TBK1 in KRAS-G12D/ K128R mutant cells could be attributed to enhanced RALB activity (Fig. 5G). We next assessed whether TBK1 activation in KRAS-G12D/ K128R mutant cells is NF1-dependent. We found that suppression of *NF1* led to increased TBK1 phosphorylation in SW1990 expressing KRAS-G12D, whereas *NF1* depletion abrogated the difference in TBK1 activity between SW1990 cells expressing KRAS-G12D or KRAS-G12D/ K128R mutants (Fig. 5H; Appendix Fig. S9C). These results indicate that the ubiquitination of the KRAS-G12D mutant at K128 suppresses the RALB-TBK1 branch of the RAS signaling pathway in an NF1-dependent manner.

The upregulation of related IκB kinase homologs, TBK1 and inhibitor of nuclear factor kappa B kinase subunit epsilon (IKKε), activates an autocrine cytokine circuit that leads to the pro-inflammatory senescence-associated secretory phenotype (SASP) (Birch and Gil, 2020). In agreement with these observations, increased TBK1 activity in the K128R knock-in SW1990 single-cell clones led to increased expression of the SASP markers, such as Interleukin 6 (IL6), C-X-C motif chemokine ligand 1 (CXCL1), C-C motif chemokine ligand 5 (CCL5), and matrix metalloproteinases (MMPs) (Fig. 5I). Moreover, the GSEA analysis revealed a strong upregulation of the SenMayo gene expression signature in SW1990 cells expressing the K128R mutant (Fig. 5J).

Notably, GSEA analysis of the TCGA pancreatic carcinoma (PDAC) samples demonstrated that *NF1* alterations in PDAC were associated with the SASP signature (Fig. 5K). This pro-tumorigenic SASP phenotype is commonly observed during PDAC progression (Rielland et al, 2014). Suppression of TBK1 activity by the specific TBK1/ IKKε inhibitor Amlexanox (Reilly et al, 2013) abolished the difference in 2D colony growth between SW1990 cells expressing KRAS-G12D or KRAS-G12D/ K128R mutants (Fig. 5L; Appendix Fig. S9D), indicating that increased TBK1 activity promotes tumorigenicity of K128R mutant SW1990 cells. Altogether, these results demonstrate that the dysregulation of K128 ubiquitination could promote oncogenic KRAS-driven tumorigenesis by favoring a TBK1-induced SASP phenotype.

## Discussion

Increasing evidence indicates that the reversible ubiquitination of RAS-like GTPases at specific lysine residues could dramatically affect their interaction landscape in a site-specific manner (Magits and Sablina, 2022). Our in silico and in vitro studies indicate that the ubiquitination of NRAS and KRAS at K128 promotes the recruitment

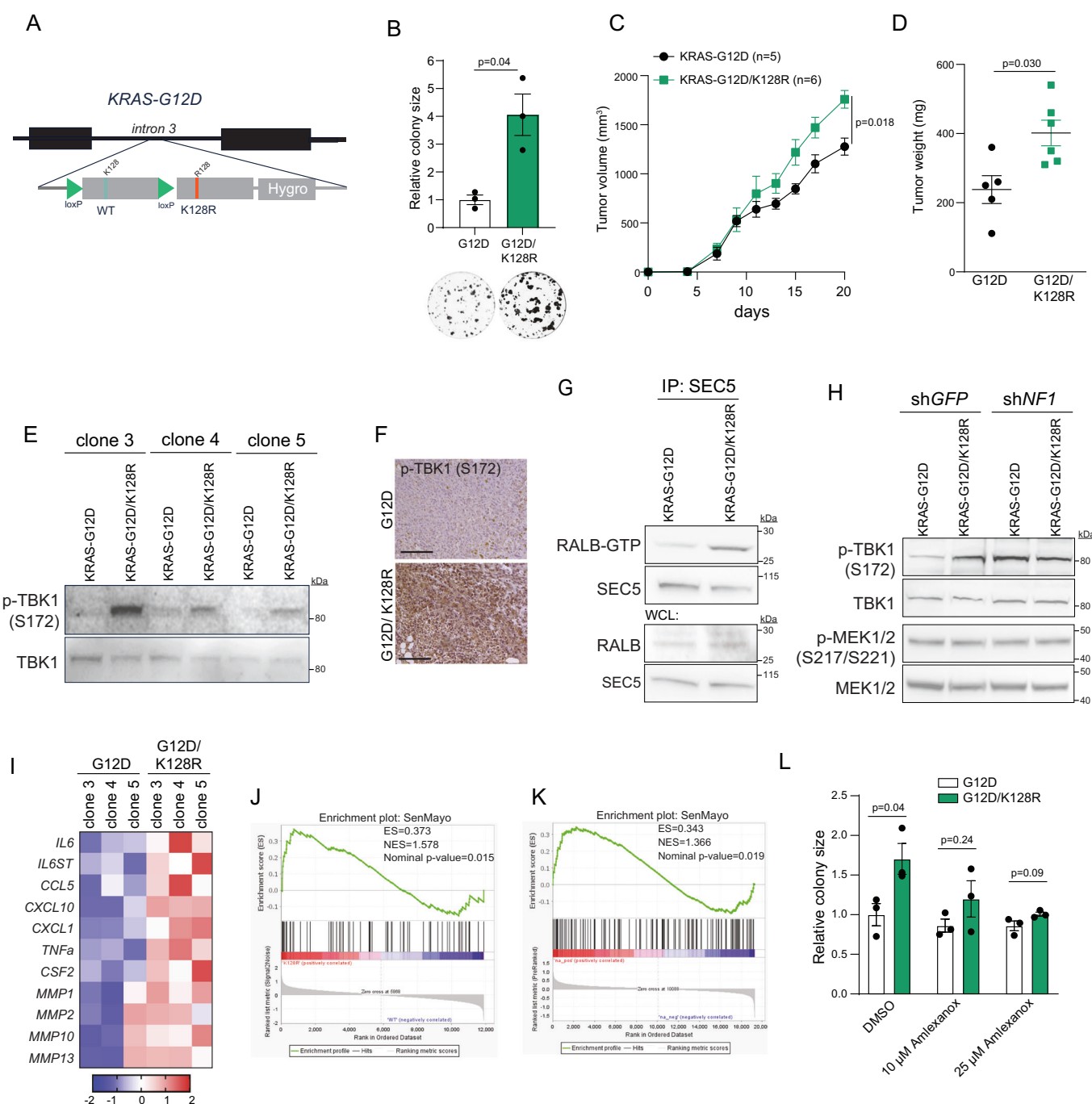

of GAP proteins, such as NF1 and RASA1, through the formation of an additional interface between ubiquitin and the extended GAP domain. Although we observed similar outcomes for the recruitment of RASA1 and NF1 to K128-ubiquitinated RAS proteins, the modes of interactions are quite different. In contrast to the majority of characterized ubiquitin-binding interfaces, the identified contacts of RASA1 with ubiquitin are in proximity to the I36 patch and the flexible β1-β2 loop of the C-terminal tail of the ubiquitin, which is similar to the interaction of VPS9 with ubiquitin (Prag et al, 2003). Moreover, the binding interfaces mainly consist of electrostatic interactions rather than hydrophobic contacts. On the other hand, the

NF1-ubiquitin contact sites closely align with the I44 hydrophobic patch of ubiquitin that controls many ubiquitin interactions (Magits and Sablina, 2022). Thus, ubiquitin seems to adopt different conformations in these two models, suggesting some degree of divergence between different GAP proteins.

Ubiquitination within the G-domain of RAS proteins presents an additional layer of the regulation of RAS activity and may provide novel approaches to target RAS-driven diseases (Campbell and Philips, 2021). Our mass-spectrometry-based studies revealed that K128 is the major site of NRAS and KRAS ubiquitination. In contrast to the monoubiquitination of RAS proteins at lysines 104, 117, or 147, which

**Figure 5.  Dysregulated ubiquitination of KRAS-G12D at K128 promotes activation of the RAL-TBK1 branch.**

(A) Schematic illustration of the CRISPR-Cas9-based approach to generate conditional K128R knock-in SW1990 clones. (B) 2D colony formation assay of three independent SW1990 single-cell clones expressing either KRAS-G12D or KRAS-G12D/K128R. Data were present as mean ± s.e.m. $N = 3$ biological replicates. P-value was calculated by a two-sided t-test. (C) Tumor growth of SW1990 xenografts expressing either KRAS-G12D or KRAS-G12D/ K128R. Xenograft volumes were evaluated for 20 days every two days. Data were presented as mean ± s.e.m.; the P-value was determined by two-way ANOVA. $N = 5$–6 biological replicates. (D) Tumor weight of SW1990 xenografts expressing either KRAS-G12D or KRAS-G12D/ K128R mutants at the endpoint. Data were presented as mean ± s.e.m.; the P-value was determined by a two-sided t-test. $N = 5$–6 biological replicates. (E) Immunoblot analysis of phosphorylated and total TBK1 expression in SW1990 single-cell clones expressing either KRAS-G12D or KRAS-G12D/K128R. (F) Immunohistochemistry analysis of phosphorylated TBK1 (p-TBK1) in SW1990 xenografts expressing either KRAS-G12D or KRAS-G12D/ K128R. Scale bar, 100 μm. (G) RALB activity in SW1990 cells expressing either KRAS-G12D or KRAS-G12D/K128R. The RALB activity was determined by its ability to interact with SEC5. (H) Immunoblot analysis of phosphorylated and total TBK1 expression in the indicated SW1990 cells expressing shGFP or shNF1. (I) The SASP-related gene expression in three SW1990 single-cell clones expressing either KRAS-G12D or KRAS-G12D/K128R. (J) GSEA analysis of the gene expression profiles in SW1990 single-cell clones expressing KRAS-G12D or KRAS-G12D/K128R. $N = 3$ biological replicates. Statistical analysis was performed by permutation test. (K) GSEA analysis of the TCGA PDAC tumors stratified by NF1 status. Statistical analysis was performed by permutation test. (L) Colony formation assay of SW1990 cells expressing either KRAS-G12D or KRAS-G12D/K128R treated with DMSO or Amlexanox. Data were present as mean ± s.e.m. $N = 3$ technical replicates in independent experiments. P-value was calculated by a two-sided t-test. Source data are available online for this figure.

stimulates the activity of RAS proteins, ubiquitination at K128 suppresses RAS signaling (Campbell and Philips, 2021). Decreased levels of K128 ubiquitination in lung squamous cell carcinoma compared to normal tissue support the tumor suppressive role of this protein modification. Whereas stimuli for RAS ubiquitination at other lysine residues are not known, the ubiquitination of K128 is transiently induced in response to cytokine and/or growth factor stimulation. Increased levels of KRAS ubiquitination at K128 were also observed in CD4[+] T-cells after T-cell receptor stimulation and in A20 B-cells after their stimulation with anti-IgG (Dybas et al, 2019; Satpathy et al, 2015). The ubiquitination-dependent control of the extent of RAS activation might be important to avoid cell growth inhibition or senescence triggered by the high-strength activity of the RAS proteins.

The ubiquitination of mutant KRAS-G12D at K128 suppresses the activation of the RALGDS-RALB-TBK1 branch of the RAS pathway in an NF1-dependent manner. A previous study demonstrated that RALGDS competes with NF1 for RAS binding (Kikuchi et al, 1994). This suggests that a higher affinity of ubiquitinated KRAS to NF1 might switch the balance towards the NF1/ KRAS-ubiquitin complexes, leading to the suppression of RAL-TBK1 activity. TBK1 activation, together with IKKε triggered by oncogenic KRAS mutants, promotes the autocrine cytokine circuit (Zhu et al, 2014). The upregulation of cytokine production and the autocrine circuit in KRAS-mutant cells could explain the increased ubiquitination of oncogenic KRAS mutants (Xu et al, 2010). Moreover, cytokine-induced ubiquitination of KRAS at K128 could serve as a component of a negative feedback loop blocking the KRAS-driven autocrine cytokine circuit.

Recent studies reported that NF1 loss accelerates KRAS-G12D-driven tumorigenesis, indicating that NF1 restrains the oncogenic capabilities of mutant KRAS (Ramakrishnan et al, 2022; Wang et al, 2019). Our data suggest that loss of NF1 function might contribute to KRAS-driven tumorigenesis by inducing TBK1 activity and a pro-tumorigenic SASP (Cruz et al, 2019; Rielland et al, 2014). Notably, the loss of the other RAS-GAP protein, DAB2IP has been shown to promote KRAS-driven tumorigenesis by inducing TBK1 activity (Miller et al, 2023). It is also worth investigating whether KRAS ubiquitination is getting dysregulated during cancer progression, to identify mechanisms of this dysregulation and its contribution to RAS-driven cancers. Exploring KRAS's susceptibility to K128 monoubiquitination could serve as a target to develop a molecular glue or PROTAC, that would promote

degradation through K48-poly-ubiquitination. Elucidating these mechanisms will shed light on the complex regulation of RAS signaling in health and disease and may provide an emerging avenue in cancer therapy research.

## Methods

### Expression vectors and antibodies

All plasmids and shRNA constructs are listed in Appendix Table S1. Plasmids were purchased from Addgene unless stated otherwise. ShRNA constructs were purchased from Sigma-Aldrich. All cloned constructs were verified by Sanger sequencing (EurofinsGenomics).

All antibodies used in this study are listed in Appendix Table S1.

### Cell culture and reagents

SW1990 cells were purchased from ATCC (CRL-2172) and were authenticated by STR profiling (ATCC) within the last 3 years. HEK293T and HeLa cells were cultured in DMEM medium (GIBCO) supplemented with 10% fetal bovine serum and 1% penicillin/streptomycin. SW1990 cells were cultured in RPMI medium (GIBCO) supplemented with 10% fetal bovine serum and 1% penicillin/streptomycin.

Mouse embryo fibroblasts (MEFs) were prepared from C57BL/6 E13.5 embryos as described in (Steklov et al, 2018). All procedures involving animals were performed in accordance with the guidelines of the IACUC of KU Leuven and approved in project application P143/2016.

Cells were tested with a mycoplasma testing kit (Lonza MycoAlert, LT07-318) every 2 weeks. Transient transfections were performed using GeneJuice (Millipore) or Lipofectamine 3000 (Thermo Fisher). Lentiviral infections were performed as described by the RNAi Consortium (TRC). Infected cells were selected by treatment with 2 μg/ml puromycin (InvivoGen) or 400 μg/ml hygromycin-B (Thermo Fisher).

Amlexanox was purchased from InvivoGen.

### Generation of conditional CRISPR knock-in clones

KRAS-K128R and NRAS K128R/ K135R knock-in cell lines were generated using a non-homologous end joining (NHEJ)-based

CRISPR/Cas9 approach as described with minor adaptations. HeLa or SW1990 cells were co-transfected with a target-specific gRNA sequence and a frame selector sequence cloned into the pX330-CAS9 backbone and a donor template plasmid using Lipofectamine 3000 (Appendix Table S1). The donor templates were designed similarly to (Thakur and Welford, 2020) (Figs. 4D, 5A). Transfected cells were cultured for one week before starting hygromycin selection. After selection, single cells were sorted using a BD FACS Aria III Cell Sorter (BD Biosciences) and screened by PCR and Sanger sequencing using target-specific primers (Appendix Table S1; Appendix Fig. S7).

To remove the wild-type *RAS* fragment, single-cell clones were transduced with adenoviral Cre (Ad5CMV-Cre; Carver College of Medicine) at an MOI of 10. Expression of Cre and wild-type *RAS* was monitored by RT-qPCR.

## In vivo tumor xenografts

All xenograft experiments were approved and performed according to the guidelines of the Ethical Committee for Animal Experimentation of KU Leuven (118/2023). About $5 \times 10^5$ cells were resuspended in 200 µl of PBS and Matrigel (Corning) in a 1:1 ratio and were subcutaneously injected into both flanks of immunodeficient *NMRI-Foxn1^{nu/nu}* mice (Janvier Labs, females, 10-week old). Mice were kept under specific pathogen-free conditions at the KU Leuven Animal Research Facility. Tumor volumes were monitored every 2 days, using a caliper until 20 days after injection. The experiment was terminated once tumors exceeded 2000 mm$^3$.

## Gene expression analysis

RNA extraction was performed using a RNeasy MINI kit (Qiagen). About 500 ng of total RNA was used to prepare cDNA for mRNA analysis using RT2 First Strand Kit. The quantification of mRNA was performed on Light Cycler 480 (Roche) system using RT2 SYBR Green qPCR Mastermix. Used primers are listed in Appendix Table S1.

RNAseq was performed using the Illumina NovaSeq 6000 sequencing platform using 150 bp paired-end reads. The overall quality of the resulting fastq files was checked using FastQC and filtered for adapter contamination using CutAdapt. Pre-processed reads were mapped to the human reference transcriptome using GENCODE (v44) and quantified using the RSEM software package. Genes with an average FPKM <1 were filtered out of the expression matrix. Raw and processed data were deposited to the GEO database with accession number GSE261976.

PDAC and LUSC datasets were obtained from the Cancer Genome Atlas (TCGA) and the Clinical Proteomic Tumor Analysis Consortium (CPTAC) data portal (Edwards et al, 2015). Samples were stratified by either NF1 status or KRAS K128 ubiquitination level. GSEA was performed using a preranked gene list based on log2 fold change (GSEA v4.3.2 software). The number of permutations was set at 1000.

## Purification of recombinant proteins, chemical ubiquitination, and in vitro assays

The recombinant proteins used: RASA1 Human Recombinant Protein (Full-length, Abnova, P01, NP_072179.1) and the GDP/ GTP exchange domain of SOS1 (564-1049aa, Cytoskeleton). Flag-NRAS WT(1-166aa), Flag-NRAS-C118S/ K128C (1-166aa), His-Ubiquitin-G76C and GST-NF1^GRD (1198-1530aa) were purified from *E. coli* as described in (Baker et al, 2013a). GST-RASA1^GAP (719-1137aa) was expressed and purified using the TNT® SP6 High-Yield Wheat Germ Protein Expression System (Promega) as described by the manufacturer's protocol.

Disulfide bond formation between Ubiquitin-G76C and NRAS-C118S/ K128C was performed with minor modifications as described in (Baker et al, 2013a). To purify NRAS-K128C conjugated to Ubiquitin-G76C, the reaction mixture was subjected to two consecutive rounds of size-exclusion chromatography using HiLoad 16/600 Superdex 75 pg column (GE Healthcare).

For in vitro immunoprecipitation experiments, NRAS was loaded with GDP or GTPγS by mixing RAS with a 10x stochiometric excess of GDP or GTPγS (Jena Bioscience) in a buffer containing an excess (20 mM) of EDTA. The immunoprecipitations were performed in the presence of 1.2 µM GDP or GTPγS.

GTPase activity was measured according to the manufacturer's protocol (Mondal et al, 2015). Briefly, equimolar amounts of NRAS or NRAS-Ub proteins were mixed with the GAP domain of RASA1 in the molar ratio 1:8. GTP hydrolysis reaction was initiated by the addition of 1.2 µM GTP and performed for 45 min at room temperature.

## Co-immunoprecipitation and immunoblotting experiments

Cells were washed in cold PBS and scraped on ice in the IP lysis buffer (50 mM Tris-HCl pH 7.5, 150 mM NaCl, 1% NP-40) (Thermo Scientific) containing protease inhibitor and phosphatase inhibitor cocktails (Roche). Cell lysates were centrifuged for 10 min at 16,000 × g at 4 °C, and proteins were immunoprecipitated using anti-Flag (M2) (Sigma-Aldrich) overnight at 4 °C. Proteins were eluted with 3×Flag (Sigma-Aldrich).

For immunoprecipitation of ubiquitinated proteins, HEK293T cells were co-transfected with 6xHis–ubiquitin and Flag–N/KRAS. Ubiquitinated proteins were purified as described previously (Simicek et al, 2013). Briefly, cells were lysed in a co-immunoprecipitation buffer containing an EDTA-free protease inhibitor cocktail (Roche). Cell lysates were mixed with His-buffer A (PBS, pH 8.0, 6 M guanidinium-HCl, 0.1% NP-40, and 1 mM β-ME) and added to TALON beads (Clontech). After binding, the resin was washed with His-buffer B (PBS, pH 8.0, 0.1% NP-40, 5% glycerol, 20 mM imidazole).

For TAP purification, Flag-tagged RAS proteins were immuno-precipitated using anti-Flag (M2) agarose (Sigma-Aldrich), washed twice with lysis buffer, once with buffer containing 50 mM Tris, at pH 7.5, 100 mM LiCl, and eluted with 3xFlag peptide. Ubiquitinated Flag-tagged RAS was further purified using TALON beads (Clontech). MS analysis was performed as previously described in (Simicek et al, 2013).

## Ubiquitome analysis

Cells were washed twice with PBS and scraped in urea lysis buffer containing 9 M urea, 20 mM HEPES pH 8.0, 1 mM sodium orthovanadate, 2.5 mM sodium pyrophosphate, 1 mM ß-glycerophosphate.

Samples were sonicated and centrifuged for 15 min at $16,000 \times g$ at room temperature. About 12 mg of total protein per sample was used for further analysis. Proteins in each sample were reduced by incubating with 4.5 mM DTT for 30 min at 55 °C. Alkylation of the proteins was done by the addition of 10 mM iodoacetamide for 15 min at room temperature. The samples were diluted with 20 mM HEPES pH 8.0 to a urea concentration of 4 M and the proteins were digested with lysyl endopeptidase (Wako) for 4 h at room temperature. All samples were further diluted with 20 mM HEPES pH 8.0 to a final urea concentration of 2 M and proteins were digested with 120 µg trypsin (Promega) overnight at 37 °C.

Immunocapture of GG-modified peptides was then performed using the PTMScan® Ubiquitin Remnant Motif (K-ε-GG) Kit (Cell Signaling Technology) according to the manufacturer's instructions. Briefly, peptides were purified on Sep-Pak C18 cartridges (Waters), lyophilized for 2 days, and re-dissolved in the immunoprecipitation buffer. Aliquots corresponding to 100 µg of digested protein material were taken for shotgun proteomics analysis. Peptides were incubated with the antibody-bead slurry for 2 h on a rotator at 4 °C. GG-modified peptides were eluted in 100 µl 0.15% TFA and desalted on reversed phase C18 OMIX tips (Agilent). Purified GG-modified peptides were dried under vacuum in HPLC inserts and stored at −20 °C until LC-MS/MS analysis.

Purified GG-modified peptides were re-dissolved in 20 µl loading solvent A (0.1% TFA in water/ACN (98:2, v/v)), of which 10 µl was injected for LC-MS/MS analysis on an Ultimate 3000 RSLCnano system in-line connected to a Q Exactive HF mass spectrometer (Thermo Scientific). From the aliquots for shotgun proteomics analysis, ~3 µg of peptides were injected for LC-MS/MS analysis on an Ultimate.

3000 HPLC system in-line connected to an LTQ Orbitrap Elite mass spectrometer (Thermo Scientific). Data analysis was performed with MaxQuant (version 1.5.7.4) using the Andromeda search engine with default search settings, including a false discovery rate set at 1% on both the peptide and protein levels. Two different searches were performed to analyze the spectra from the GG-enriched samples and the shotgun samples separately. In both searches, spectra were interrogated against all mouse proteins in the Uniprot/SwissProt database (www.uniprot.org). The minimum score for modified peptides was set to 30 and for the GlyGly-enriched samples, GlyGly modification of lysine residues was set as an additional variable modification.

Further data analysis was performed with the Perseus software (version 1.5.5.3) after loading the GlyGly (K) Sites table from MaxQuant. Reverse database hits were removed as well as potential contaminants. Sites with less than three valid values in at least one group were removed and missing values were imputed from a normal distribution around the detection limit.

## Protein stability

Protein stability in living cells was assessed by the Global Protein Stability (GPS) approach (Najm et al, 2021; Yen et al, 2008). Twenty-four hours after plating, HEK293T cells were transfected with the GPS reporter. Forty-eight hours after transfection, cells were harvested and analyzed using the MACSQuant VYB Flow Cytometer (Miltenyi Biotec). Raw data were analyzed using FlowJo Software (BD Biosciences). Live single cells were monitored for the expression of GFP and DsRed, and the GFP/ DsRed ratio was counted to measure the relative protein stability.

## Immunostaining

For immunofluorescence, $2 \times 10^4$ cells were plated on eight-well chamber slides (Ibidi) and fixed with 4% PFA 24 h after transfection. Cells were permeabilized in PBS-0.1% Triton-X100 and blocked with 5% BSA. Primary anti-HA antibody (Roche, 3F10) and secondary Alexa568-conjugated anti-rat antibody (Life Technologies) were applied by diluting in blocking buffer before mounting in Vectashield antifade mounting medium (VectorLabs). Confocal images were obtained using a Leica SPII microscope (Leica Microsystems, Wetzlar, Germany).

For immunohistochemistry, tumors were fixed in 4% paraformaldehyde in PBS and embedded in paraffin. Paraffin slides were rehydrated and treated with hydrogen peroxide. Antigen retrieval was performed by heat in Tris-EDTA pH 9.0. The sections were incubated with primary and secondary antibodies and diaminobenzidine (Dako) was used as a detection method followed by hematoxylin counterstaining.

## Split-CAT-based ubiquitin-binding assays

The assays were performed as previously described in (Levin-Kravets et al, 2021). E. coli Mach1 was co-transformed with pC-CAT-Ubiquitin and pN-CAT-RASA1$^{GAP}$ or pN-CAT-NF1$^{GRD}$ vectors and selected on agar plates supplemented with 30 µg/ml kanamycin and 25 µg/ml streptomycin respectively. Double resistance colonies grew in liquid LB, spotted at even amounts on agar plates supplemented with 8 µg/ml chloramphenicol, and growth at 37 °C was monitored for 36 h using an office scanner. The growth was quantified as described in (Levin-Kravets et al, 2016).

## In silico modeling

The initial structure of ubiquitin was obtained from the Protein Data Bank (PDB: 1UBQ), and the initial coordinates of NRAS and KRAS were obtained from our previous works (Jang et al, 2020; Steklov et al, 2018). In the modeling, ubiquitin was conjugated to K128 of RAS using the patch parameter for the isopeptide bond connecting ubiquitin to the lysine residue. Six ubiquitinated NRAS systems were generated to ensure that ubiquitin covers all possible interfaces at the allosteric lobe of NRAS. To construct the ubiquitinated NRAS/ RASA1$^{GAP}$ complex, all configurations of simulated ubiquitinated NRAS were superimposed on the crystal structure of HRAS in complex with RASA1$^{GAP}$ (PDB: 1WQ1). We collected one of the best initial configurations of the ubiquitinated NRAS/ RASA1$^{GAP}$ complex based on the contact between ubiquitin and RASA1 for the all-atom simulations. To construct the ubiquitinated KRAS/ NF1$^{GRD}$ complex, the crystal structure of the non-ubiquitinated KRAS/ NF1$^{GRD}$ complex (PDB: 6V65) was used to construct the ubiquitinated systems. Four different initial configurations of the ubiquitinated KRAS/ NF1$^{GRD}$ complex with the initial ubiquitin positions taken from the ubiquitinated NRAS systems were generated for the all-atom simulations.

We performed MD simulations using the updated CHARMM all-atom force field (version 36 m) to construct the set of starting points and relax the systems to a production-ready stage. Our simulations followed closely the same protocol as in our previous

work (Grudzien et al, 2022; Jang et al, 2021; Liu et al, 2023). The TIP3 water model was used to solve the system. Sodium (Na+) and chlorine (Cl−) ions were added to create a final ionic strength of 150 mM and neutralize the system. In the pre-equilibrium stage, a series of minimization and dynamics cycles were performed for the solvents, including the ions, with the harmonically restrained protein backbone until the solvent reached 310 K. The harmonic restraints on the protein backbones were progressively removed through the dynamics cycles using the long-range particle mesh Ewald (PME) electrostatics calculation. Production runs were performed using the NAMD parallel computing code on a Biowulf cluster at the National Institutes of Health (Bethesda, MD). In the production runs, the Langevin thermostat maintained the constant temperature at 310 K and the Nosé-Hoover Langevin piston pressure control sustained the pressure at 1 atm with the NPT condition. The SHAKE algorithm was applied to constrain the motion of bonds involving hydrogen atoms. A 2 fs timestep for 1 μs was used for all simulations. The CHARMM program was used to analyze the simulation trajectories. In the analysis, ensemble clustering was implemented in Chimera to obtain the most populated conformational representatives.

## Statistical analysis

No blinding was employed in experiments. The error bars indicate the standard error of the mean (s.e.m.). All data were analyzed using GraphPad Prism.

# Data availability

The plasmids generated in this studies available upon request. RNAseq of SW1990 single-cell clones is available at GEO accession GSE261976.

The source data of this paper are collected in the following database record: biostudies:S-SCDT-10_1038-S44318-024-00146-w.

# Peer review information

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

## Acknowledgements

This work was supported by the H2020 European Research Council (ub-RASDisease, ID: 772649) (to AS). This project has been funded in whole or in part with federal funds from the National Cancer Institute, National Institutes of Health, under contract HHSN261201500003I. The content of this publication does not necessarily reflect the views or policies of the Department of Health and Human Services, nor does mention of trade names, commercial products, or organizations imply endorsement by the US Government. This research was supported [in part] by the Intramural Research Program of the NIH, National Cancer Institute, Center for Cancer Research (to MZ, HJ, and RN), and by Israel Science Foundation 1440/21 and Israel Cancer Research Fund 940283 (to GP). The research was supported [in part] by the Grant Agency of the Czech Republic (GA CR 22-26981 S) (to MS). All simulations were performed using the high-performance computational facilities of the Biowulf PC/Linux cluster at the National Institutes of Health, Bethesda, MD (https://hpc.nih.gov/). We thank Prof. Francis and Dr. Delphi Van Haver (VIB Proteomics Core) for performing ubiquitome analysis.

## Author contributions

**Wout Magits**: Data curation; Formal analysis; Investigation; Visualization; Methodology; Writing—original draft; Writing—review and editing. **Mikhail Steklov**: Conceptualization; Formal analysis; Supervision; Validation; Investigation; Visualization; Methodology; Writing—review and editing. **Hyunbum Jang**: Investigation; Methodology; Writing—review and editing. **Raj N Sewduth**: Investigation; Methodology; Writing—review and editing. **Amir Florentin**: Investigation; Methodology; Writing—review and editing. **Benoit Lechat**: Investigation; Methodology; Writing—review and editing. **Aidana Sheryazdanova**: Investigation; Methodology; Writing—review and editing. **Mingzhen Zhang**: Investigation; Methodology; Writing—review and editing. **Michal Simicek**: Funding acquisition; Investigation; Methodology; Writing—review and editing. **Gali Prag**: Supervision; Funding acquisition; Methodology; Writing—review and editing. **Ruth Nussinov**: Supervision; Funding acquisition; Methodology; Writing—review and editing. **Anna Sablina**: Conceptualization; Data curation; Supervision; Funding acquisition; Visualization; Writing—original draft; Project administration; Writing—review and editing.

Source data underlying figure panels in this paper may have individual authorship assigned. Where available, figure panel/source data authorship is

listed in the following database record: biostudies:S-SCDT-10_1038-S44318-024-00146-w.

## Disclosure and competing interests statement

The authors declare no competing interests.

