## [Peer Review File · The EMBO Journal]

K128 ubiquitination constrains RAS activity by expanding the binding interface with GAP proteins

Wout Magits, Mikhail Steklov, Hyunbum Jang, Raj Sewduth, Amir Florentin, Benoit Lechat, Aidana Sheryazdanova, Mingzhen Zhang, Michal Simicek, Gali Prag, Ruth Nussinov, and Anna Sablina

Corresponding author(s): Anna Sablina (Anna.Sablina@cme.vib-kuleuven.be)

Review Timeline:

Submission Date:	17th Oct 23
Editorial Decision:	24th Oct 23
Appeal Received:	24th Oct 23
Editorial Decision:	23rd Jan 24
Revision Received:	3rd Apr 24
Editorial Decision:	10th May 24
Revision Received:	13th May 24
Accepted:	29th May 24

Editor: Ieva Gailite

Transaction Report:

Dear Dr. Sablina,

Thank you for submitting your manuscript "K128 ubiquitination constrains RAS activity by expanding the binding interface with GAP proteins" to The EMBO Journal. I sincerely apologise for the delay in the assessment of your manuscript due to the high number of submissions that we receive at the moment. I have now read your study carefully and discussed the work with other members of the editorial team. I regret to inform you that we have decided not to pursue the publication at The EMBO Journal, but I would like to suggest a transfer to our sister journal EMBO Reports.

We appreciate that your describes the role of KRAS and NRAS mono-ubiquitination at K128 for activation of their GTPase activity via increased interaction with RAS GAPs. The findings further show that K128 mono-ubiquitination suppresses downstream KRAS and NRAS signalling pathway activation both in serum-activated MEFs and HeLa cells and in pancreatic cancer cells expressing mutant KRAS-G12D. Finally, K128 ubiquitination of KRAS is reduced in lung adenocarcinoma patient samples and induces senescence-associated secretory phenotype.

While we appreciate the novelty of the presented characterisation of KRAS/NRAS K128 mono-ubiquitination, we also find that further identification of either the E3 ligase mediating this activation or analysis of its in vivo role, e.g., using mouse xenografts, would be needed at this point. Therefore, we unfortunately concluded that your manuscript in its current form is not a sufficiently strong candidate for publication in The EMBO Journal.

That being said, I appreciate the novelty of the findings and have discussed the manuscript with my colleague Achim Breiling at our sister journal EMBO Reports. I am glad to say that he would like to send your study out for peer review. If you are interested in this option, please use the transfer link below. Please note that no reformatting of the manuscript is necessary for a transfer to EMBO Reports.

Link Not Available

Thank you for giving us the opportunity to consider this manuscript. I am sorry that I cannot offer better news this time, and I hope that you will find the transfer option of interest.

Yours sincerely,

Ieva Gailite

** As a service to authors, EMBO Press provides authors with the possibility to transfer a manuscript that one journal cannot offer to publish to another EMBO publication or the open access journal Life Science Alliance launched in partnership between EMBO Press, Rockefeller University Press and Cold Spring Harbor Laboratory Press. The full manuscript and if applicable, reviewers' reports, are automatically sent to the receiving journal to allow for fast handling and a prompt decision on your manuscript. For more details of this service, and to transfer your manuscript please click on Link Not Available. **

Dear Dr. Galite,

Thank you for your feedback.

I am wondering if we add the xenograft experiments, you will be more enthusiastic to consider the manuscript? The experiment for the PDAC model is currently ongoing.

Best regards,

Anna Sablina

Dear Dr. Sablina,

Thank you for your letter regarding the recent decision on your manuscript. If you were to add PDAC experiments and if the outcome is in line with the experiments in the cell lines, I would be happy to reconsider the manuscript at The EMBO Journal. In this case, I would recommend to indicate in the cover letter that it is a resubmission of the manuscript EMBOJ-2023-115909, so that the two papers can be linked.

With best regards,

leva

Dear Dr. Sablina,

Thank you for submitting your manuscript for consideration by the EMBO Journal. I sincerely apologise for the protracted review process for your manuscript due to delays in referee report submission. We have now received comments from two reviewers, which are included below for your information.

As you will see from the reports, both reviewers find the study per se of interest, while also pointing out a number of important aspects that would need to be addressed in the final version. In particular, both reviewers indicate that the analysis of TBK1 activity regulation by RAS K128 ubiquitination needs to be extended. Based on the interest expressed in the reports, I would like to invite you to address the issues raised by the referees in a revised manuscript. I think it would be useful to discuss the revision in more detail via email or phone/videoconferencing - please let me know which option you prefer.

We generally allow three months as standard revision time. As a matter of policy, competing manuscripts published during this period will not negatively impact on our assessment of the conceptual advance presented by your study. However, please contact me as soon as possible upon publication of any related work to discuss the appropriate course of action. Should you foresee a problem in meeting this three-month deadline, please contact us to arrange an extension.

When preparing your letter of response to the referees' comments, please bear in mind that this will form part of the Review Process File and will therefore be available online to the community. For more details on our Transparent Editorial Process, please visit our website: <https://www.embopress.org/page/journal/14602075/authorguide#transparentprocess>. Please also see the attached instructions for further guidelines on preparation of the revised manuscript.

Please feel free to contact me if you have any further questions regarding the revision. Thank you for the opportunity to consider your work for publication. I look forward to discussing your revision.

With best wishes,

Ieva

Ieva Gailite, PhD
Senior Scientific Editor
The EMBO Journal
Meyerohofstrasse 1
D-69117 Heidelberg
Tel: +4962218891309
i.gailite@embojournal.org

We realize that it is difficult to revise to a specific deadline. In the interest of protecting the conceptual advance provided by the work, we recommend a revision within 3 months (22nd Apr 2024). Please discuss the revision progress ahead of this time with the editor if you require more time to complete the revisions. Use the link below to submit your revision:

Referee #1:

Regulation of RAS function by post-translational modifications (PTMS) is an understudied but important aspect of RAS regulated signaling. Here, Magits et al. attempt to define the functional role of K128 ubiquitination (Ub) in the cellular regulation of NRAS and KRAS activity. Using a hybrid approach involving molecular modelling and dynamics, molecular biology, and bioinformatics, they claim that K128-Ub occurs in response to growth factor stimulation to attract GAPs (i.e. RASA1 and NF1) and limit the duration of RAS activity. Furthermore, they claim the mechanism promotes tumorigenesis by mutant KRAS to promote a SASP phenotype driven by TBK1 phosphorylation.

The first claim, that K128-Ub inhibits RAS activity through attraction is nicely described using MD, modeling, and pulldown experiments. However, while the MD/modelling experiments were put in the context of truncated crystal structures of RAF, NF1, and RASA1, they should be placed in the context of the full-length structures, as many of these proteins now have cryo-em or alpha-fold analogues that could be used to provide these analyses.

The second major claim, that both K128-Ub and NF1 loss facilitate TBK1 phosphorylation to drive a SASP phenotype is exceedingly weak. For instance, only one of the engineered clones is used for the biological experiments is worrisome. More needs to be done for these analyses, including testing alternative hypotheses, like PI3K/AKT/mTORC signaling, that could explain the SASP, colony formation, tumor and histology phenotypes observed in the mutant cells. More extensive molecular modeling may help with this. A knockdown experiment of TBK1 in the engineered cell lines and its effect on colony formation would be useful to support their model.

Overall, the model and mechanism are novel and provide insight into feedback inhibition by GAPs. However, the overall data produced to support their model and mechanism is sparse and more experiments are needed. In particular, part of the model involving TBK1 and the SASP phenotype is weak and requires more data for support. In my opinion the novelty of the model and mechanism presented are within the scope of the journal if the problems/criticisms presented here are addressed. Besides the general criticisms above, I have provided more specific comments below:

1. In general, the western blots are faint. Is it possible to make these darker?
2. No discussion is given for the distribution of NRAS/KRAS ubiquitination? For instance, is there a possible reason why is K117 Ub absent in KRAS? A plot of HRAS PTMs would be useful to include in Supp fig. 1.
3. Why NRAS and KRAS, but not HRAS, in characterization experiments of Figure 1-3? A justification would be useful, or better yet inclusion of these experiments in the paper.
4. Why does the Ub-blot in S1C not show change in bands? Shouldn't they match the anti-flag blot to some degree?
5. "whereas substituting either K117 or K147 led to decreased degradation of KRAS, substituting K128 did not affect KRAS stability (Fig.1C)" This statement is incorrect as a double mutant, and not the individual mutations, were used in Fig.1C.
6. "the cycloheximide pulse-chase experiment analysis revealed that the NRAS-K128R/K135R mutant exhibited a similar degradation rate compared to wt-NRAS (Fig.S21,B)." Without a positive control to determine if the pulse-chase worked, it is hard to believe this statement.
7. The modelling and simulations are good. However, full-length structures or relevant multimeric complex structures are now available for RAS effectors/regulators like RAF and NF1 (e.g. cryo-EM, Alpha-fold, hybrid models). The authors should put the ubiquitinated modeling and MD simulations in the context of these structures. These analyses may help address the broader issue of alternative hypotheses to explain the biology of K128-Ub.
8. "In the pull-down experiments, K128 monoubiquitinated GTP-loaded NRAS..." Should be GTP-analogue loaded NRAS.
9. The SOS binding experiment in Fig. 2E lacks a control.
10. Are the experiments in Fig. 2 done with active site mutant RAS? What are the nucleotide bound states in Fig. 2F? The binding of GAP should promote hydrolysis and subsequent release of the GAP protein. If the pulldown experiments were allowed to incubate overnight, even at 4°C, then reduction in hydrolysis by K128-Ub seems very plausible. An alternative

interpretation of data in Fig.2F and G is that 128-Ub promotes nucleotide exchange, increases the GTP or GTP-analogue bound state of KRAS, and thereby increases the pulldown of RAS and apparent amount of GTP hydrolyzed. A more rigorous kinetic analysis of the RAS-Ub + GAP system on GTP hydrolysis (such as in Hobbs et al. 2019 PMID: 31649109) would be useful to determine if K128-Ub promotes an increase in GTP hydrolysis through a change in affinity. Also, in the materials and methods section under GTPase activity assay 'equimolar amounts' means that NRAS and GAP were combined at a ratio of 1:1, rather than 1:8.

11. Why do the Split-CAT assay rather than IPs? Can RASA1 or NF1 bind Ub on their own in an IP assay?

12. Are the changes in Fig.4A,B meaningful? In particular, NRAS which appears to show little change over 30 minutes after serum application.

13. Unlike Figure 4F, and Supp Fig.6C,D shows no effect of RASA1/NF1 knockout. I believe this point needs to be addressed quantitatively.

14. Akt signaling appears different in clone 5 of Supplementary figure 7A, in contrast to statement that connects it to the manuscript. Also, only one clone in Fig. 5C appears to have significantly altered TBK1 phosphorylation.

15. What about the other clones in fig. 5H,I,J? Do these clones show the same result? Also, a simple knockdown of TBK1 in the clonogenic assay would strengthen the argument for that TBK1 mediates the biological effect of K128-Ub/NF1-loss.

16. Fig. 5K lacks identification of the clones/mutation in the panels.

17. Could the authors provide sequencing data in the supplement of the KRAS gene for the clones in Fig. 5 to ensure that only the correct site was mutated, as well as the zygosity of the alteration?

Referee #2:

The manuscript by Anna Sablina and colleagues describes a posttranslational modification of the ras oncogene that may regulate its interaction with GTPase-activating proteins (GAPs), thereby leading to altered downstream signalling that is relevant in cancer biology.

In particular, the authors mapped a ubiquitination modification to residue K128 that appears to have a regulatory role rather than causing degradation. The authors propose an interesting mechanism in that K128-ubiquitinated RAS enhances the interaction surface for GAP, NF1 and RASA1, which increases RAS binding to GAP proteins and promotes GAP-mediated GTP hydrolysis. Most importantly, the authors show that K128 ubiquitylation of the KRAS-G12D mutant form contributes to tumour growth by interfering with the RAL1/TBK1 pathway.

Generally, there has been a great interest in RAS biology and its tumour variants as oncogenes, especially in the advent of selective inhibitors having reached the clinic.

In this context, the present study represents some novel findings regarding KRAS (mono)ubiquitination that creates an expanded interaction surface leading to functional consequences that are worth reporting. However, some concerns need to be addressed before this manuscript can be recommended for publication.

Specific comments (major & minor concerns):

1. Figure 1. What is the dynamics between the different mono-ubiquitylation events on RAS, such as K104, K117 and K128, during starvation and short-term activation? Do these events happen simultaneously or are mutually exclusive?

Western blot from this and other studies would suggest a RAS-mono-ubiquitylated form to be predominant at steady-state, but this may change dependent on physiological conditions. This maybe relevant related to the dynamics of activating downstream pathways.

2. Figure 5C shows that the selected single-clones SW1990 cells expressing either KRAS-G12D or KRAS-G12D/K128R exhibit considerable heterogeneity regarding TBK1/TBK1 phosphorylation (S172) profiles.

How generalisable is then the claim that the TBK1 signalling axis is perturbed when you interfere with K128 ubiquitylation? Have the authors screened a wider range of single knockout clones to put this claim on a more solid basis?

3. Technical issues - western blot gels do not show molecular weight markers (except in Figure 1). They should be included (Figures 2D,E,F,H,K, 3C,D,G, 4D,E,F and 5C,D).

4. Some language syntax errors that should be amended.

5. Have the authors explored which E3 ubiquitin ligase that performs RAS K128-ubiquitylation? Cullin 3 ?

We would like to thank the reviewers for the essential and constructive suggestions that helped us to considerably improve the manuscript.

Reviewer 1:

1. In general, the western blots are faint. Is it possible to make these darker?

We avoided oversaturated exposure of blots to maintain the band signal intensity in a linear range according to published guidelines (Janes, 2015, Tie *et al.*, 2021).

2. No discussion is given for the distribution of NRAS/KRAS ubiquitination. For instance, is there a possible reason why is K117 Ub absent in KRAS?

We do not have a clear explanation for the differences in ubiquitination of NRAS and KRAS at K117 and K147. The observed difference or lack of KRAS ubiquitination at 117 could not be fully explained by detection limitations, as we identified multiple peptides that corresponded to these sites (revised Fig S1D). KRAS ubiquitination at K117 was also not detected in a previous study (Sasaki *et al.*, 2011). Even though we observed a different distribution of NRAS and KRAS ubiquitination at K117 and K147, the ubiquitination at K128 was the most prevalent modification for both NRAS and KRAS. We have discussed this point in the revised manuscript.

A plot of HRAS PTMs would be useful to include in Supp fig. 1

We have added the profile of post-translational modifications of HRAS to the revised Fig S1C.

3. Why NRAS and KRAS, but not HRAS, in characterization experiments of Figure 1-3? A justification would be useful, or better yet inclusion of these experiments in the paper.

As HRAS presents arginines at positions 128 and 135, it could not be ubiquitinated at these sites. For this reason, HRAS was not included in further experiments. We have clarified this in the revised manuscript.

4. Why does the Ub-blot in S1C not show change in bands? Shouldn't they match the anti-flag blot to some degree?

The ubiquitin immunoblot served as a control for the pull-down of ubiquitinated proteins present in the lysate. With this control, we can exclude that the difference between wild-type NRAS and NRAS mutants originates from differences in the His-pull-down efficiency between the samples. On the other hand, Flag immunoblot revealed ubiquitination of NRAS.

5. "whereas substituting either K117 or K147 led to decreased degradation of KRAS, substituting K128 did not affect KRAS stability (Fig.1C)" This statement is incorrect as a double mutant, and not the individual mutations, were used in Fig.1C.

We thank the reviewer for identifying this inaccuracy. As the ubiquitination of RAS at K128 is the major focus of our manuscript, we decided to remove the stability data for KRAS-K117Q/K147L from the revised manuscript.

6. "the cycloheximide pulse-chase experiment analysis revealed that the NRAS-K128R/K135R mutant exhibited a similar degradation rate compared to wt-NRAS (Fig.S21,B)." Without a positive control to determine if the pulse-chase worked, it is hard to believe this statement.

To confirm the results of the cycloheximide pulse-chase experiments, we have performed a global stability assay (GPS) using wild-type NRAS and NRAS-K128R/K135R mutant. In agreement with the cycloheximide pulse-chase experiments, we observed that K128R/K135R mutations did not affect NRAS stability (revised Fig 1C). Similarly, the GPS assay did not reveal any difference between wild-type KRAS and KRAS-K128R mutant (Fig 1C), suggesting a non-degradative role of K128 ubiquitination. To maintain consistency in the experimental setup for KRAS and NRAS, we decided to remove the cycloheximide pulse-chase experiment from the manuscript.

7. The modelling and simulations are good. However, full-length structures or relevant multimeric complex structures are now available for RAS effectors/regulators like RAF and NF1 (e.g. cryo-EM, Alpha-fold, hybrid models). The authors should put the ubiquitinated modeling and MD simulations in the context of these structures. These analyses may help address the broader issue of alternative hypotheses to explain the biology of K128-Ub.

We performed modeling in the context of cryoelectron microscopy (cryo-EM) structures of the full-length NF1 homodimer (Chaker-Margot et al, 2022). The GAP-related domain of NF1 (NF1^{GRD}) is exposed for RAS binding only in the open conformation. The modeling of K128-ubiquitinated KRAS with the open conformation of the NF1 homodimer showed that ubiquitin could form the interaction only with the NF1^{GRD} domain (revised Fig. S4). Therefore, we performed MD simulations on the K128-ubiquitinated KRAS in a complex with the GAP-related domain of NF1 (NF1^{GRD}) (Fig 3E, Fig S5).

8. "In the pull-down experiments, K128 monoubiquitinated GTP-loaded NRAS..." Should be GTP-analogue loaded NRAS.

We thank the reviewer for identifying this inaccuracy. We have now corrected this error in the revised manuscript.

9. The SOS binding experiment in Fig. 2E lacks a control.

In Figure 2E, we compare the ability of recombinant ubiquitinated and non-ubiquitinated NRAS to bind recombinant SOS1. We controlled both the loading of all

the proteins and the efficacy of the pull-down of ubiquitinated and non-ubiquitinated NRAS.

10. Are the experiments in Fig. 2 done with active site mutant RAS? What are the nucleotide bound states in Fig. 2F? The binding of GAP should promote hydrolysis and subsequent release of the GAP protein. If the pulldown experiments were allowed to incubate overnight, even at 4°C, then reduction in hydrolysis by K128-Ub seems very plausible. An alternative interpretation of data in Fig.2F and G is that 128-Ub promotes nucleotide exchange, increases the GTP or GTP-analogue bound state of KRAS, and thereby increases the pulldown of RAS and apparent amount of GTP hydrolyzed. A more rigorous kinetic analysis of the RAS-Ub + GAP system on GTP hydrolysis (such as in Hobbs et al. 2019 PMID: 31649109) would be useful to determine if K128-Ub promotes an increase in GTP hydrolysis through a change in affinity.

All in vitro experiments were performed using wild-type RAS proteins. For the RASA1^{GAP} interaction experiment, we used NRAS or NRAS-Ub preloaded with a non-hydrolyzable GTP γ S analog. The experiment was also performed in the presence of 1.2 μ M GTP γ S. Thus, the observed increased interaction between NRAS-Ub and RASA1^{GAP} could not be explained by alterations in GTP hydrolysis or GDP/GTP exchange rate. We have clarified the details of in vitro experiments in the revised Material and Methods and Figure Legends sections.

We agree with the reviewer that the kinetic analysis of GTP hydrolysis could be useful to further prove that ubiquitination at K128 leads to increased RASA1-induced GTP hydrolysis due to a change in affinity to RASA1. However, the limited amount of the purified NRAS-Ub did not allow us to perform this assay.

Also, in the materials and methods section under GTPase activity assay 'equimolar amounts' means that NRAS and GAP were combined at a ratio of 1:1, rather than 1:8.

We used equimolar amounts of NRAS and NRAS-Ub, and the ratio of NRAS or NRAS-Ub to RASA1^{GAP} was 1 to 8. We have clarified this in the revised Material and Methods.

11. Why do the Split-CAT assay rather than IPs? Can RASA1 or NF1 bind Ub on their own in an IP assay?

Ubiquitin interacting motifs bind mono-ubiquitin with a low affinity, ranging from 2 μ M to 2,100 μ M (Prag *et al.*, 2003, Ren & Hurley, 2010), posing a challenge for conventional biochemical and biophysical studies. The Split-CAT system allows the sensing of low-affinity ubiquitin-binding domains (Levin-Kravets *et al.*, 2016).

12. Are the changes in Fig.4A,B meaningful? In particular, NRAS which appears to show little change over 30 minutes after serum application.

We observed a dynamic alteration in RAS ubiquitination at K128 in response to serum stimulation (Fig 4A,B) as shown by an increase at 5 minutes and a subsequent decline to baseline levels 30 minutes after the stimulation. MS-based analysis and immunoblotting showed similar results (Fig 4A,B). A similar transient increase of RAS ubiquitination at K128 was observed upon B-cell receptor (BCR) stimulation (Satpathy *et al.*, 2015)(revised Fig. S6A). Of note, ubiquitination at K147 showed only a slight increase, and no alteration of ubiquitination of NRAS at 117 was detected in response BCR stimulation (Fig. S6A). In our ubiquitome analysis, other sites were not detected. Such a dynamic regulation of RAS ubiquitination points out its role in a negative feedback loop mechanism in response to stimuli.

13. Unlike Figure 4F, and Supp Fig.6C,D shows no effect of RASA1/NF1 knockout. I believe this point needs to be addressed quantitatively.

We found a consistent effect of RASA1 or NF1 knockout on MAPK signaling in different experiments (Fig 4F and Fig S6C,D, now revised Fig S8C,D) that was confirmed using the quantification of the immunoblots (rebuttal Fig 1).

Figure 1 Quantification of MEK1/2 phosphorylation levels. The immunoblots were quantified using ImageJ. Values are levels of phosphorylated MEK1/2 relative to total protein levels.

14. Akt signaling appears different in clone 5 of Supplementary figure 7A, in contrast to statement that connects it to the manuscript. Also, only one clone in Fig. 5C appears to have significantly altered TBK1 phosphorylation.

We agree with the reviewer that the K128R mutation differentially affected the activity of AKT signaling in different clones (Fig S9B). On the other hand, the KRAS-G12D/K128R mutant led to consistent activation of the TBK1 pathway in all three single-cell clones (Fig 5E). The difference in the level of TBK1 activation between clones could be due to the differences between single-cell clones and/or the efficiency of Cre recombination in the distinct clones. In agreement with this idea, we found that the level of TBK1 phosphorylation correlates with the efficiency of Cre recombination in

the clones (Fig 5E; Fig S9A). Moreover, we observed a significant increased expression of TBK1-regulated cytokines (Zhu *et al.*, 2014), including IL6, CCL5, and CXCL10, in all three independent clones (Fig 5I). We have clarified this in the revised manuscript.

15. What about the other clones in fig. 5H,I,J? Do these clones show the same result? Also, a simple knockdown of TBK1 in the clonogenic assay would strengthen the argument for that TBK1 mediates the biological effect of K128-Ub/NF1-loss.

The colony formation was performed using three independent clones (revised Fig 5B). While all three clones showed increased 2D growth of SW1990 cells expressing KRAS-K128R mutant, Clone 3, which presented the highest Cre recombination efficiency, was selected for further experiments (Fig S9A).

We also found that inhibition of TBK1 by Amlexanox abolished the observed growth advantage in the KRAS-G12D/ K128R cells, further strengthening the idea that TBK1 mediates the observed phenotype. We have added this data to the revised manuscript (Fig 5L).

16. Fig. 5K lacks identification of the clones/mutation in the panels.

We apologize for this mistake. We have properly labeled the panels in the revised Fig 5F.

17. Could the authors provide sequencing data in the supplement of the KRAS gene for the clones in Fig. 5 to ensure that only the correct site was mutated, as well as the zygosity of the alteration?

We have included a full description of the validation of the generated clones in the revised Fig S7. First, we confirmed the incorporation of the cassette using PCR analysis as depicted in Fig S7A. The PCR analysis demonstrated the absence of non-edited alleles in the generated clones. We also confirmed the incorporation of the cassette using the Sanger sequencing (Fig S7B).

Reviewer 2:

1. Figure 1. What is the dynamics between the different mono-ubiquitylation events on RAS, such as K104, K117 and K128, during starvation and short-term activation? Do these events happen simultaneously or are mutually exclusive? Western blot from this and other studies would suggest a RAS-mono-ubiquitylated form to be predominant at steady-state, but this may change depending on physiological conditions. This may be relevant related to the dynamics of activating downstream pathways.

The detection constraints of the ubiquitome analysis do not allow us to fully address this question. In our ubiquitome analysis of serum-stimulated cells, we were able to detect only NRAS and KRAS ubiquitination at K128. The ubiquitinome analysis of B-cell receptor activation (Satpathy *et al.*, 2015) revealed that whereas NRAS and KRAS

ubiquitination at K128 transiently increased upon anti-IgG stimulation, ubiquitination at K147 showed only a slight increase, and no alteration of ubiquitination of NRAS at 117 was detected (Fig S6A). Moreover, whereas we observed decreased levels of NRAS and KRAS ubiquitination at K128 in the CPTAC lung tumour samples compared to normal tissue, ubiquitination at K117 was not altered and ubiquitination at other sites was detected (Fig S6B). This suggests that RAS ubiquitination at different sites is regulated by distinct mechanisms.

2. Figure 5C shows that the selected single-clones SW1990 cells expressing either KRAS-G12D or KRAS-G12D/K128R exhibit considerable heterogeneity regarding TBK1/TBK1 phosphorylation (S172) profiles. How generalizable is then the claim that the TBK1 signalling axis is perturbed when you interfere with K128 ubiquitylation? Have the authors screened a wider range of single knockout clones to put this claim on a more solid basis?

The K128R mutation led to consistent activation of the TBK1 pathway in three independent clones. The difference in TBK1 activation between the clones could be due to the differences between single-cell clones or the efficiency of Cre recombination in the distinct clones. In agreement with this idea, we observed that the level of TBK1 phosphorylation correlates with the efficiency of Cre recombination in the clones (Fig 5E, Fig S9A). Moreover, we observed increased expression of TBK1-regulated cytokines (Zhu *et al.*, 2014), including IL6, CCL5, and CXCL10, in all three independent clones (Fig 5I). We have clarified this in the revised manuscript.

3. Technical issues - western blot gels do not show molecular weight markers (except in Figure 1). They should be included (Figures 2D,E,F,H,K, 3C,D,G, 4D,E,F and 5C,D).

We have added weight markers to the revised figures.

4. Some language syntax errors that should be amended.

We diligently addressed and fixed the errors in the updated version of the manuscript.

5. Have the authors explored which E3 ubiquitin ligase that performs RAS K128-ubiquitylation? Cullin 3 ?

Even though the identification of the ubiquitin machinery responsible for NRAS and KRAS ubiquitination at K128 is important, it is out of the scope of our manuscript. Previous studies demonstrated that the MLN4924 inhibitor, which blocks the activity of Cullin-RING ubiquitin ligases (CRL) dramatically decreased levels of RAS ubiquitination (Steklov *et al.*, 2018). Given the prevalence of ubiquitination at K128, this suggests the contribution of CRL in the control of RAS ubiquitination at K128.

References:

Janes, K. A. (2015). *Sci Signal* **8**, rs2.

- Levin-Kravets, O., Tanner, N., Shohat, N., Attali, I., Keren-Kaplan, T., Shusterman, A., Artzi, S., Varvak, A., Reshef, Y., Shi, X., Zucker, O., Baram, T., Katina, C., Pilzer, I., Ben-Aroya, S. & Prag, G. (2016). *Nat Methods* **13**, 945-952.
- Prag, G., Misra, S., Jones, E. A., Ghirlando, R., Davies, B. A., Horazdovsky, B. F. & Hurley, J. H. (2003). *Cell* **113**, 609-620.
- Ren, X. & Hurley, J. H. (2010). *EMBO J* **29**, 1045-1054.
- Sasaki, A. T., Carracedo, A., Locasale, J. W., Anastasiou, D., Takeuchi, K., Kahoud, E. R., Haviv, S., Asara, J. M., Pandolfi, P. P. & Cantley, L. C. (2011). *Sci Signal* **4**, ra13.
- Satpathy, S., Wagner, S. A., Beli, P., Gupta, R., Kristiansen, T. A., Malinova, D., Francavilla, C., Tolar, P., Bishop, G. A., Hostager, B. S. & Choudhary, C. (2015). *Mol Syst Biol* **11**, 810.
- Steklov, M., Pandolfi, S., Baietti, M. F., Batiuk, A., Carai, P., Najm, P., Zhang, M., Jang, H., Renzi, F., Cai, Y., Abbasi Asbagh, L., Pastor, T., De Troyer, M., Simicek, M., Radaelli, E., Brems, H., Legius, E., Tavernier, J., Gevaert, K., Impens, F., Messiaen, L., Nussinov, R., Heymans, S., Eyckerman, S. & Sablina, A. A. (2018). *Science* **362**, 1177-1182.
- Tie, L., Xiao, H., Wu, D.-l., Yang, Y. & Wang, P. (2021). *Acta Pharmacologica Sinica* **42**, 1015-1017.
- Zhu, Z., Aref, A. R., Cohoon, T. J., Barbie, T. U., Imamura, Y., Yang, S., Moody, S. E., Shen, R. R., Schinzel, A. C., Thai, T. C., Reibel, J. B., Tamayo, P., Godfrey, J. T., Qian, Z. R., Page, A. N., Maciag, K., Chan, E. M., Silkworth, W., Labowsky, M. T., Rozhansky, L., Mesirov, J. P., Gillanders, W. E., Ogino, S., Hacohen, N., Gaudet, S., Eck, M. J., Engelman, J. A., Corcoran, R. B., Wong, K. K., Hahn, W. C. & Barbie, D. A. (2014). *Cancer Discov* **4**, 452-465.

Dear Anna,

Thank you for submitting a revised version of your manuscript. I sincerely apologise for the protracted assessment process due to the high number of submissions we receive at the moment.

Your study has now been seen by both original referees, who now find that most of their previous concerns have been addressed and broadly recommend acceptance of the manuscript.

There now remain a few editorial points that need addressing before I can extend acceptance of the manuscript:

1. Please submit up to five keywords.
2. Please submit a complete author checklist, which you can download from our author guidelines (<https://www.embopress.org/pb-assets/embo-site/EMBO%20Press%20Author%20Checklist-1642513524327.xlsx>). Please insert information in the checklist that is also reflected in the manuscript. The completed author checklist will also be part of the Review Process File.
3. Please check that the funding information is correct and identical both in the manuscript and our online system. Intramural Research Program of the NIH, National Cancer Institute, Center for Cancer Research (to RN), and by Israel Science Foundation 1440/21 are currently missing in our online system.
4. Please move "References" section before Figure legends.
5. Please add a "Disclosure and competing interests statement" section after "Acknowledgments" (further info: <https://www.embopress.org/page/journal/14602075/authorguide#conflictsofinterest>).
6. CRediT has replaced the traditional author contributions section because it offers a systematic, machine-readable author contributions format that allows for more effective research assessment. Please remove the Authors Contributions from the manuscript and use the free text boxes beneath each contributing author's name in our online submission system to add specific details on the author's contribution. More information is available in our guide to authors.
7. In the Appendix, please adjust the nomenclature to "Appendix Figure S1" etc. Table S1 should be added to the Appendix PDF and nomenclature should be corrected to "Appendix Table S1". Please preface the Appendix with a brief table of contents.
8. In our standard source data check, we have noted unexplained duplicate values figures 1C, 5C and 5I. I have attached the corresponding files with the detected duplications labelled in colour. Please take a look and correct as needed. A brief explanation would be very helpful.
9. Our data editors have flagged the following issues in figure legends that need correcting:
 - The legend for figure 4g-i is mislabeled as 4f, h, g, respectively - please correct.
 - The figure panels for supplementary figures 9c-d is incorrectly labelled as 9d, c.
 - Please indicate the statistical test used for data analysis in the legends of figures 4d, h-i; 5j-k.
 - Please define the box plots in terms of minima, maxima, centre, bounds of box and whiskers, and percentile in the legends of figures 3a, e; 4g, supplementary figure 6b.
 - Please add information on the number and nature of replicates in the legends of figures 2b, j; 3a, e; 4d; 5c.
 - Please describe the nature of replicates in the legends of figures 1c; 2g; 3b, f; 4a; 5b, d, l, supplementary figures 8a-b.
 - Please note define the error bars in the legends of figures 2b, j; 4d.
 - Please note that in figure 1d the scale bar unit should be corrected from μM to μm (in the figure legend).

Finally, papers published in The EMBO Journal are accompanied online by a 'Synopsis' to enhance discoverability of the manuscript. It consists of A) a short (1-2 sentences) summary of the findings and their significance, B) 3-4 bullet points highlighting key results and C) a synopsis image that is 550x300-600 pixels large (width x height, jpeg or png format). You can either show a model or key data in the synopsis image. Please note that the image size is rather small and that text needs to be readable at the final size. Please send us this information together with the revised manuscript.

Thank you again for giving us the chance to consider your manuscript for The EMBO Journal. I look forward to receiving the final version and your input on the source data issues.

With best wishes,

leva

leva Gailite, PhD
Senior Scientific Editor
The EMBO Journal
Meyerhofstrasse 1
D-69117 Heidelberg

Tel: +4962218891309
i.gailite@embojournal.org

We realize that it is difficult to revise to a specific deadline. In the interest of protecting the conceptual advance provided by the work, we recommend a revision within 3 months (8th Aug 2024). Please discuss the revision progress ahead of this time with the editor if you require more time to complete the revisions. Use the link below to submit your revision:

Referee #1:

Regulation of RAS function by post-translational modifications (PTMS) is an important aspect of RAS regulation that is understudied. Magits et al. define the functional role of K128 ubiquitination (Ub) in the regulation of NRAS and KRAS. Using a hybrid approach involving molecular modelling and dynamics, molecular biology, and bioinformatics, they claim that K128-Ub occurs in response to growth factor stimulation to attract GAPs (i.e. RASA1 and NF1) and limit the duration of RAS activity. They show that Ub acts to downregulate TBK1 function through KRAS to promote tumorigenesis via SASP. Furthermore, they speculate that induction of SASP by Ub-KRAS suppression of TBK1 phenocopies NF1 loss. The authors do a good job describing the functionality of Ub-RAS and its impact on cell signaling, and the speculative role that it plays in tumorigenesis is certainly interesting.

The authors address all my concerns, with exception of the following, but I believe this concern is minor.

Point 4 from reviewer 1.

I still find this set of blots problematic. While I acknowledge that the Ub blot reflects universal 6x-His-Ub proteins, shouldn't the total N-Ras blot show changes in migration with the addition of ~8.5 kDa and ~17kDa in response to Ubiquitination, as it does in the anti-flag blot? Perhaps indicate in the legend if different types of polyacrylamide were used (e.g. gradient versus single percentage polyacrylamide gel).

Referee #2:

This is a revised version of the manuscript describing the role of K128 ubiquitylation on RAS biological function. This study is novel and of relevance to the RAS oncology field.

The authors have addressed previous concerns raised by the reviewers including the addition of revised and new data. The scientific parts are now sound. This reviewer would recommend publication pending some minor text revisions:

- Page 12, bottom two lines
...by favouring a GEOa TBK1-induced SASP phenotype.

- Page 13, line 18
Our mass-spectroscopy-based studies
(should be)
Our mass-spectrometry-based studies

All editorial and formatting issues were resolved by the authors.

Dear Anna,

Thank you for addressing the final editorial points. I am now pleased to inform you that your manuscript has been accepted for publication. Congratulations on a nice study!

Before we forward your manuscript to our publishers, I would like to propose some minor edits in the manuscript title, abstract and synopsis (please see below and the attached manuscript text file). I have also written a short blurb that will accompany the title of your manuscript in our online system. Please take a look and let me know if any corrections are needed:

Title:

K128 ubiquitination constrains RAS activity by expanding its binding interface with GAP proteins

Blurb:

Reduction of K128 mono-ubiquitination activates wild-type and mutant RAS signaling and elicits a senescence-associated secretory phenotype, promoting RAS-driven tumorigenesis.

Synopsis

RAS family protein ubiquitination at several residues regulates RAS signaling activity and stability. This study shows that mono-ubiquitination of KRAS and NRAS at lysine 128 (K128) enhances RAS binding to GTPase-activating proteins to restrict RAS activity, a mechanism that is dysregulated in cancer tissues.

- K128 monoubiquitination of KRAS or NRAS creates an additional interface that facilitates binding to the GTPase activating proteins RASA1 and NF1.
- The level of KRAS and NRAS ubiquitination at K128 is decreased in cancer samples compared to normal tissue.
- K128 ubiquitination of wild-type RAS proteins constrains downstream MAPK signalling in a GAP-dependent manner.
- K128 ubiquitination of the oncogenic KRAS-G12D mutant suppresses the RAL-TBK1 axis and inhibits the autocrine circuit induced by mutant KRAS.

If you have any questions, please do not hesitate to contact the Editorial Office. Thank you for this contribution to The EMBO Journal and congratulations on a successful publication!

With best wishes,

Ieva

Ieva Gailite, PhD
Senior Scientific Editor
The EMBO Journal
Meyerohofstrasse 1
D-69117 Heidelberg
Tel: +4962218891309
i.gailite@embojournal.org

>>> Please note that it is The EMBO Journal policy for the transcript of the editorial process (containing referee reports and your response letter) to be published as an online supplement to each paper. If you do NOT want this, you will need to inform the Editorial Office via email immediately. More information is available here: <https://www.embopress.org/transparent->

process#Review_Process
